# One for All: Towards Training One Graph Model for All Classification Tasks

**Hao Liu**[1*]    **Jiarui Feng**[1*]    **Lecheng Kong**[1*]    **Ningyue Liang**[1]    **Dacheng Tao**[2]
**Yixin Chen**[1]    **Muhan Zhang**[3†]
[1]Washington University in St. Louis    [2] Nanyang Technological University    [3]Peking University
{liuhao, feng.jiarui, jerry.kong, fliang, ychen25}@wustl.edu,
dacheng.tao@ntu.edu.sg, muhan@pku.edu.cn

## Abstract

Designing a single model to address multiple tasks has been a long-standing objective in artificial intelligence. Recently, large language models have demonstrated exceptional capability in solving different tasks within the language domain. However, a unified model for various graph tasks remains underexplored, primarily due to the challenges unique to the graph learning domain. First, graph data from different areas carry distinct attributes and follow different distributions. Such discrepancy makes it hard to represent graphs in a single representation space. Second, tasks on graphs diversify into node, link, and graph tasks, requiring distinct embedding strategies. Finally, an appropriate graph prompting paradigm for in-context learning is unclear. We propose **One for All (OFA)**, the first general framework that can use a single graph model to address the above challenges. Specifically, OFA proposes text-attributed graphs to unify different graph data by describing nodes and edges with natural language and uses language models to encode the diverse and possibly cross-domain text attributes to feature vectors in the same embedding space. Furthermore, OFA introduces the concept of nodes-of-interest to standardize different tasks with a single task representation. For in-context learning on graphs, OFA introduces a novel graph prompting paradigm that appends prompting substructures to the input graph, which enables it to address varied tasks without fine-tuning. We train the OFA model using graph data from multiple domains (including citation networks, molecular graphs, knowledge graphs, etc.) simultaneously and evaluate its ability in supervised, few-shot, and zero-shot learning scenarios. OFA performs well across different tasks, making it the first general-purpose across-domains classification model on graphs.

## 1 Introduction

Recently, large language models (LLMs) have received tremendous attention due to their power and versatility in solving natural language tasks like text generation, machine translation, and question-answering. LLMs' in-context learning ability and universality allow the model to directly perform various cross-domain downstream tasks by providing related context or prompt to the model, therefore avoiding any fine-tuning on model parameters (Brown et al., 2020; Zhang et al., 2023b; Lu et al., 2021; Bommasani et al., 2021).

Despite the great success of foundation models on language, developing a foundation model for graph structure data is less explored. Particularly, several challenges unique to graph data prevent the direct transfer of foundation model design from the language domain to the graph domain. First, although the natures of language tasks differ, they are still uniformly represented in human-interpretable texts. An LLM can encode them into the same text embedding space and train on different source tasks together. However, **graph datasets from different sources are usually completely different in feature representation**. Concretely, widely used graph datasets include citation

---

*Contributed equally. Listing order is random.
†Corresponding author.

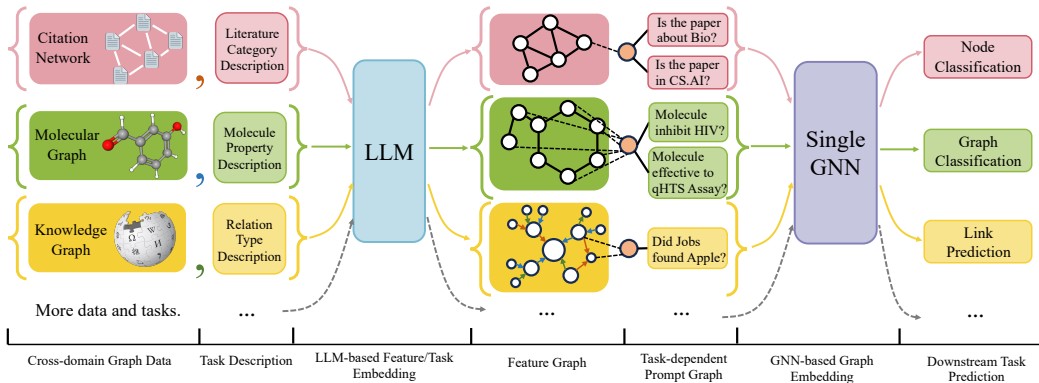

Figure 1: The pipeline of OFA. An input to the model contains a text-attributed graph and a task description. Cross-domain texts in graphs and task descriptions can be co-embedded in the same space by an LLM. OFA's graph prompting paradigm converts the input with embedded features to prompted graphs with a unified task representation, which allows adaptive downstream prediction.

networks (Yang et al., 2016; Hu et al., 2020), e-commerce networks (Shchur et al., 2018), knowledge graphs (Dettmers et al., 2018; Toutanova & Chen, 2015), and molecular graphs (Dwivedi et al., 2020). Their raw forms contain attributes generated from isolated processes. For example, node features in molecular graphs are usually vectors whose entries are indices of nominal features of atoms. In contrast, node features in e-commerce networks could be Bag-of-Word vectors of item descriptions. These features are so different in dimension, scale, and semantic meanings that it is almost impossible to directly learn the representation of these data using the same model. Second, **different downstream tasks in the graph domain attend to different parts of the graph and require task-specific knowledge and methodologies**. Specifically, graph-related tasks can be roughly divided into three classes: node-level, link-level, and graph-level. Even though Graph Neural Networks (GNNs) achieved great success in all three task classes, the rationale for the success behind each task class is different. For node-level tasks, proper smoothing of the node features leads to good performance (Defferrard et al., 2016; Chien et al., 2021; He et al., 2021). However, for link-level and graph-level tasks, encoding the local structure is vital to the success, encouraging a line of work that develops more expressive GNNs (Xu et al., 2018; Zhang & Li, 2021; Zhang & Chen, 2018). Generally, a powerful model for node-level tasks may not work on link-level or graph-level tasks. Consequently, current models are incompetent and infeasible to learn different tasks jointly. Third, the design of the in-context learning or prompt is straightforward in natural language, where we can simply add a description of the task or a few examples to the input. However, there is no existing solution to add such context information to the graph *generically*. **How to design a unified way to perform cross-domain and in-context learning on the graph tasks is ambiguous.**

To address these challenges, we propose **One-for-All (OFA)**, a general solution for building and training a foundation GNN model with in-context learning ability across different domains. OFA has three main unique features: (1) OFA uses text-attributed graphs (TAGs) to integrate graph datasets from different domains into one large TAG dataset and leverages the power of LLMs to learn from all domains jointly. We collect nine graph datasets commonly used in the community varying in size, domains, and task types (see Table 1 for the full list). Then, we describe all nodes and edges in the graphs using human-readable texts and embed the texts from different domains into the same embedding space with a single LLM. (2) OFA proposes the nodes-of-interest (NOI) subgraph and the NOI prompt node, which not only unify different types of graph tasks but also improve the ability of the foundation model to learn the structural information in the graph. (3) OFA introduces a carefully designed and widely applicable graph prompting paradigm (GPP) that inserts a prompt graph into the original input graph in a task-specific way. The nodes in the prompt graph contain all related information about the downstream task (described by texts and encoded by the same LLM encoder as the input graph). Then, the modified graph becomes the actual input to the foundation graph model. Thus, the model is adaptive to perform different tasks according to the prompt graphs. Figure 1 illustrates the pipeline of OFA. **After training, the users can describe any graph with natural texts and apply the OFA pipeline to predict possibly unseen classes.**

Table 1: Detailed summarization of all collected datasets in OFA.

| Dataset | Domain | Task | # Graphs | Avg. #Nodes | Avg. #Edges | # Classes |
|---------|--------|------|----------|-------------|-------------|-----------|
| Cora | Citation | Node/Link | 1 | 2,708 | 10,556 | 7 |
| PubMed | Citation | Node/Link | 1 | 19,717 | 44,338 | 3 |
| ogbn-arxiv | Citation | Node | 1 | 169,343 | 1,166,243 | 40 |
| Wiki-CS | Web link | Node | 1 | 11,701 | 216,123 | 10 |
| FB15K237 | Knowledge | Link | 1 | 14,541 | 310,116 | 237 |
| WN18RR | Knowledge | Link | 1 | 40,943 | 93,003 | 11 |
| PCBA | Molecule | Graph | 437,929 | 26.0 | 28.1 | 128 |
| HIV | Molecule | Graph | 41,127 | 25.5 | 27.5 | 2 |
| ChEMBL | Molecule | Graph | 365,065 | 25.9 | 55.9 | 1048 |

We evaluate the proposed OFA on all collected TAG datasets under supervised, few-shot, and zero-shot scenarios. We demonstrate that a single OFA model can perform well on cross-domain and cross-task scenarios. In particular, we show that the OFA achieves great results on zero-shot learning, which is impossible for most of the existing graph models.

## 2 PRELIMINARIES

**Text-attributed graphs (TAGs).** We define a TAG as a graph where each node and each edge in the graph is associated with a text sentence. We denote a TAG as $\mathcal{G} = (\mathcal{V}, \mathcal{E}, \mathcal{R})$, where $\mathcal{V} = \{v_1, \ldots, v_{|\mathcal{V}|}\}$ is the set of nodes, $\mathcal{R} = \{r_1, \ldots, r_{|\mathcal{R}|}\}$ is the set of relations, $\mathcal{E} = \{e_1, \ldots, e_{|\mathcal{E}|}\}$ is the set of edges. An edge $e_{ij} = (v_i, r, v_j) \in \mathcal{E}$ consists of a source node $v_i \in \mathcal{V}$, a relation $r \in \mathcal{R}$, and a target node $v_j \in \mathcal{V}$. Let $s_{v_i}$ denote the text sentence associated with node $v_i$ and $s_{e_{ij}}$ denote the text sentence associated with edge $e_{ij} \in \mathcal{E}$. Finally, let $\mathcal{N}_k(v)$ denote the set of all neighbor nodes within $k$ hops of node $v$. Note that some concurrent works also introduce the concept of TAGs (Chen et al., 2023; He et al., 2023). However, they only focus on graphs whose raw node features are already texts. On the contrary, we extend this concept and regard all graphs as TAGs since any nodes and edges are describable by texts.

**Learning scenarios.** In this work, we focus on the classification problem. Denote a dataset as $\mathcal{D} = \{(d_i, y_i)\}_1^D$, where $D$ is the number of data in the dataset, $d_i$ is a data sample, $y_i \in \mathcal{Y}$ is the label of $d_i$ and $\mathcal{Y}$ is the set of all data labels. To train a classifier, we split the dataset into the train, validation, and test sets, denoted as $\mathcal{D}_{train}$, $\mathcal{D}_{val}$, and $\mathcal{D}_{test}$ respectively. Their label sets are $\mathcal{Y}_{train}$, $\mathcal{Y}_{val}$, and $\mathcal{Y}_{test}$. We focus on three learning scenarios. In **supervised learning**, the model will be trained on $\mathcal{D}_{train}$ and be evaluated on $\mathcal{D}_{val}$ to determine the best model. Finally, $\mathcal{D}_{test}$ is used to evaluate the model's performance. All labels in the validation and test data are seen during training, that is, $\mathcal{Y}_{train} = \mathcal{Y}_{test}$. The second learning scenario is **few-shot learning**. The training and evaluation procedure of few-shot learning is similar to supervised learning. However, in few-shot learning, we have $\mathcal{Y}_{train} \bigcap \mathcal{Y}_{test} = \emptyset$. Few-shot learning typically deals with $N$-way $K$-shot tasks, where we use $N \cdot K$ data $\{(d_i, y_i)\}_1^{N \cdot K}$ as support samples, such that each distinct class is provided with $K$ labeled data and $N$ is the total number of distinct classes. Next, given these support samples, the model needs to classify data in the query set $\mathcal{Q} = \{d_i\}_1^n$ into these $N$ classes. The third learning scenario is **zero-shot learning**, which can be viewed as a special case of few-shot learning. In zero-shot learning, for any $N$-way $K$-shot task, we have $K = 0$ as there are no support samples.

**In-context learning in language.** In-context learning mainly refers to the ability of the model to learn tasks given only a few examples in the form of demonstration (Dong et al., 2023). For the language model, this is mainly achieved by the prompting mechanism. The pretrained LLM model takes the demonstration as input, which is provided by the prompt text $C$. Then, the answer to the task is given by generating the rest of the sentence conditioned on $C$ (Brown et al., 2020).

## 3 ONE-FOR-ALL: TOWARDS FOUNDATION MODEL ON GRAPH

The proposed OFA is a general graph learning framework that uses one model to simultaneously solve classification tasks varying in formats and backgrounds, similar to LLMs that can answer substantially different questions using the same model weight. Figure 1 illustrates the pipeline of OFA. OFA can be divided into three parts. First, graphs from different domains are integrated into

text-attributed graphs with the same format, allowing a single LLM to embed all TAGs into the same space. In the second part, OFA unifies different task types in the graph domain by introducing the Nodes-of-Interest (NOI) subgraph and NOI prompt node, where a graph model can attend to task-relevant information automatically. Finally, OFA proposes the Graph Prompting Paradigm (GPP) that organically injects task information into the graph data, enabling in-context learning.

## 3.1 UNIFYING GRAPH DATA FROM DIFFERENT DOMAINS WITH TAGS

One critical challenge in building a foundation model for graphs is that cross-domain graph data are usually generated by entirely different procedures and have node/edge attributes embedded in different spaces. This makes graph models trained on one domain almost impossible to generalize to another domain. However, despite the distinct attributes across datasets, almost all can be described by human-interpretable language. For example, in molecular graphs where nodes represent atoms, we can use plain text to describe the node with atomic features, including element names, chirality, etc. The key advantage is that by using text to describe nodes and edges, we can apply an LLM to encode different graph attributes into the same space. Consequently, we introduce the concept of TAGs to integrate graph data from different domains systematically.

Specifically, we design a standardized format for text feature generation of any nodes and edges in graph data. The text feature format for nodes is shown below:

> **Text feature of nodes:** Feature node. *<feature description>*: *<feature content>*; *<feature description>*: *<feature content>*; ...
> **Example:** Feature node. Atom: Carbon, Atomic number 6, helix chirality, is not in a ring, ...
> **Example:** Feature node. Paper title and abstract: Attention is all you need. The dominant sequence transduction models are ...

Given a TAG $\mathcal{G}$, the text feature $s_{v_i}$ always starts with the text *Feature node.* to indicate that this node is an input node with features from the original graph as opposed to prompt nodes, which will be introduced in Section 3.3. Next, the text describes the type of a feature, followed by the content of the feature. If there are multiple features for a node, they are joined by semicolons. The construction of the text feature $s_{e_{ij}}$ for edge $e_{ij}$ is similar, except the start of the text is *Feature edge.*

> **Text feature of edges:** Feature edge. *<feature description>*: *<feature content>*; *<feature description>*: *<feature content>*; ...
> **Example:** Feature edge. Chemical Bond: ionic bonding, is conjugated, ...
> **Example:** Feature edge. Citation from one paper to another.

Following the protocol, we meticulously collected nine graph datasets widely recognized as benchmarks in numerous downstream tasks. This collection encompasses graph data from various domains, including citation networks, molecular graphs, knowledge graphs, and more. Additionally, it covers nearly all classification tasks employed in the research community, i.e., node classification, link prediction, and graph classification. We provide the detailed summarization of all the collected datasets in OFA in Table 1 and detailed collection and processing protocol in Appendix B.1.

As mentioned above, we can apply an LLM encoder to encode all text features into a fixed-length vector as the final input feature of all nodes/edges. Namely, for node $v_i$ and edge $e_{ij}$, their vector representations are defined as $x_i = \text{LLM}(s_{v_i})$ and $x_{ij} = \text{LLM}(s_{e_{ij}})$. Because the LLM-encoded input features contain domain information, the subsequent pipeline can capture and leverage this information. Generally, any kind of LLM can be used as the encoder, and a stronger LLM potentially yields better overall performance. In OFA, we evaluate and compare the performance of different LLMs (further discussion in Section 5). We also provide a visualization of all generated OFA datasets in Appendix B.2.

## 3.2 UNIFYING DIFFERENT GRAPH TASKS WITH NODES-OF-INTEREST

Downstream classification tasks in the graph domain can be divided into different categories like: (1) **node-level tasks**, where the task is to classify a node in the graph; (2) **link-level tasks**, where the task is to reason about the connection between a node pair; (3) **graph-level tasks**, where the task is to

make prediction on the whole graph. However, tasks at different levels need to be handled by distinct procedures and methods, which makes the construction of a foundation model for graphs difficult. In contrast, different downstream tasks in language share the same autoregressive generation nature, which makes the knowledge learned from the next-token prediction task used in LLMs uniformly beneficial to various downstream tasks. Then the question arises: Can we unify different graph tasks into a single task to facilitate the training and knowledge transferring in the graph domain?

In OFA, we propose Nodes-of-Interest (NOI) subgraph and NOI prompt node to achieve the goal. The term **NOI** refers to the set of target nodes in a task, illustrated by the blue nodes in Figure 2, and is represented as $\mathcal{T}$. NOI is not limited to the listed levels of tasks, and its size depends on the prediction target. An **NOI subgraph** is defined as the subgraph around the NOI. Denote $\mathcal{S}_h(v) = \{\mathcal{V}_v^h, \mathcal{E}_v^h, \mathcal{R}_v^h\}$ as the $h$-hop ego-subgraphs around $v$, consisting of $h$-hop neighbor nodes of $v$ and all interconnecting edges. A NOI subgraph $\mathcal{G}_h(\mathcal{T})$ combines ego-subgraphs of all nodes in NOI,

$$\mathcal{G}_h(\mathcal{T}) = \bigcup_{v \in \mathcal{T}} \mathcal{S}_h(v) = \{\bigcup_{v \in \mathcal{T}} \mathcal{V}_v^h, \bigcup_{v \in \mathcal{T}} \mathcal{E}_v^h, \bigcup_{v \in \mathcal{T}} \mathcal{R}_v^h\}. \tag{1}$$

For node-level tasks on node $v$, NOI is the node itself, so $\mathcal{T} = \{v\}$ and $\mathcal{G}_h(\mathcal{T}) = \mathcal{S}_h(v)$. For link-level tasks on node pair $(v_i, v_j)$, we have $\mathcal{T} = \{v_i, v_j\}$ and $\mathcal{G}_h(\{v_i, v_j\}) = \mathcal{S}_h(v_i) \bigcup \mathcal{S}_h(v_j)$. For graph-level tasks, NOI contains all nodes in the graph, and the NOI subgraph is $\mathcal{G}_h(\mathcal{V}) = (\mathcal{V}, \mathcal{E}, \mathcal{R})$.

Then, we define the **NOI prompt node** to unify the processing and readout procedures in different task types. The NOI prompt node is associated with a task prompt text:

> **Text feature of the NOI prompt node:** Prompt node. *<task description>*.
> **Example:** Prompt node. Graph classification on molecule properties.
> **Example:** Prompt node. Node classification on the literature category of the paper.

The text is encoded by the same LLM as other text in $\mathcal{G}$. The NOI prompt node connects to all nodes in NOI, as illustrated by double concentric circles in Figure 2. *Through message passing, the NOI prompt node summarizes information in the NOI and the task description.* We can then attach class nodes to the NOI prompt node for downstream tasks, which we will explain further in Section 3.3. While concurrent works also utilize subgraphs to unify different types of tasks (Sun et al., 2023; Liu et al., 2023c), these approaches mainly leverage the subgraph concept to transform tasks into a graph-level task, **without a NOI prompt node design**. In contrast, with the NOI prompt node, we do not require any explicit pooling mechanism, distinguishing our method from previous ones. The combination of NOI subgraph and NOI prompt nodes in our design achieves a unified readout and treatment for all node-level, link-level, and graph-level tasks. Further, the NOI prompt node connected to NOI can be viewed as a labeling trick, which uplifts the expressive power of the original graph model to better learn structural information around the NOI (Zhang et al., 2021). Moreover, the task prompt text on the NOI prompt node allows the graph model to adjust the readout parameters according to the specific task, which is not feasible in existing works.

### 3.3 GRAPH PROMPTING PARADIGM FOR GRAPH IN-CONTEXT LEARNING

One of the most fascinating properties of LLMs is their ability of in-context learning through prompting, which allows the model to perform various downstream tasks in different learning scenarios without fine-tuning. For example, in a few-shot scenario where the goal is to predict the category of a paper based on its title and abstract, we can provide LLMs with $k$ papers from each category as context and instruct the model to generate predictions based on the provided context. However, research on performing in-context learning for graphs remains relatively uncharted.

We recognize that the core principle of in-context learning involves manipulating the input data to align it with downstream tasks. Hence, we propose the Graph Prompting Paradigm (GPP) to manipulate the input graph so that the graph model can acquire task-relevant information from the input itself. Such a paradigm endows the graph model with in-context learning ability for both seen and unseen classes, enabling **zero-shot learning**. Concretely, a prompt graph, denoted as $\mathcal{P} = (\mathcal{V}_p, \mathcal{E}_p, \mathcal{R}_p)$ has **two types of nodes**. The first node type is the NOI prompt node, which we have introduced in section 3.2. Suppose we are querying a target NOI subgraph $\mathcal{G}_h^q(\mathcal{T}^q) = (\mathcal{V}_q^h, \mathcal{E}_q^h, \mathcal{R}_q^h)$, and the NOI prompt node is $p_q$. GPP adds edges between the NOI prompt node and every node in NOI, as illustrated by the dotted line in Figure 2. We denote them by $\mathcal{E}_{cross}^q =$

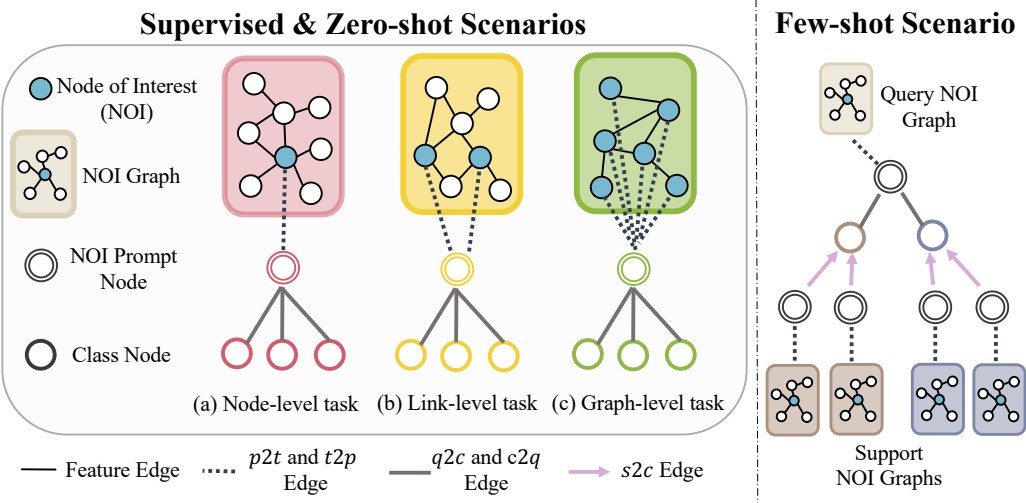

Figure 2: In-context learning design in OFA

$\{(t, r_{t2p}, p_q), (p_q, r_{p2t}, t)|t \in \mathcal{T}^q\}$. Note that $r_{t2p}$ and $r_{p2t}$ are the relation types for edges from NOI to the NOI prompt node and the reverse edges, respectively. The second node type in the prompt graph is called the **class node**. Each class node holds text information related to a specific class.

> **Text feature of class node:** Prompt node. *<class description>*.
> **Example:** Prompt node. Molecule property. The molecule is effective in: ...
> **Example:** Prompt node. Literature Category. cs.AI (Artificial Intelligence). Covers all areas of AI except Vision ...

Denote the class node for class $i$ by $c_i$. We add edges between every class node and the NOI prompt node as illustrated by the gray lines in Figure 2, denoted as: $\mathcal{E}_{query} = \{(p_q, r_{q2c}, c_i), (c_i, r_{c2q}, p_q)|i \in [N]\}$, where $N$ is the number of classes. $r_{q2c}$ and $r_{c2q}$ specify the edge relation type from the NOI prompt node to a class node and the reverse. Overall, the prompt graph $\mathcal{P} = (\mathcal{V}_p, \mathcal{E}_p, \mathcal{R}_p)$ is given by:

$$\mathcal{V}_p = \{p_q\}\bigcup\{c_i|i \in [N]\}, \mathcal{E}_p = \mathcal{E}_{query}\bigcup\mathcal{E}_{cross}^q, \mathcal{R}_p = \{r_{t2p}, r_{p2t}, r_{q2c}, r_{c2q}\}. \quad (2)$$

Then, the *prompted graph* fed to the subsequent graph model is the combination of the input graph and prompt graph, denoted as $\mathcal{G}_m = (\mathcal{V}_q^h\bigcup\mathcal{V}_p, \mathcal{E}_q^h\bigcup\mathcal{E}_p, \mathcal{R}_q^h\bigcup\mathcal{R}_p)$. We use a graph learning model to process the prompted graph and use the embeddings of the class nodes to make binary classification. Specifically, let $h_{c_i}$ be the vector representation of class node $c_i$ from the graph learning model. We predict the likelihood of the NOI belonging to class $i$ by

$$P[\text{NOI belongs to class } i] = \sigma(\text{MLP}(h_{c_i})), \quad (3)$$

where MLP is a Multi-layer Perceptron whose 1-dimensional output represents the classification score of $h_{c_i}$. Note that because the NOI prompt node and class nodes connect to the NOI and contain task text description, the fixed-length vector $h_{c_i}$ contains information about both the input graph and task, making the prediction task-dependent. While existing graph learning methods need to use different pooling mechanisms for different tasks, this formulation turns different levels of tasks into the same binary classification tasks on class nodes so all tasks can be trained together. For multi-class problems, we compare the prediction scores of different classes to make the decision.

$$l = \text{argmax}_i\left(\text{MLP}(h_{c_i})|i \in [N]\right), \quad (4)$$

$l$ is the predicted class of the NOI. Apart from the generality on task levels, because the graph model is oblivious to the class node order and can inductively infer the class based on the class text description on the class node, the users can attach arbitrary unseen class nodes with proper text descriptions to the NOI prompt node. In such cases, the model can predict the unseen class nodes based on its experience with seen class nodes whose text description is semantically close to the unseen ones, facilitating **zero-shot learning**.

The GPP can also prompt few-shot problems, where support NOI subgraphs of unseen classes are provided to help classify the query NOI subgraph better. The support NOI subgraphs are denoted by $\mathcal{G}_h^{i,k}(\mathcal{T}_k^i)$ for the $i$-th class and $k$-th support sample and the NOI $\mathcal{T}_k^i$ belong to class $i$. As for the query NOI prompt node, we connect each support NOI subgraph to its corresponding support NOI prompt node $p_{i,k}$ by

$$\mathcal{E}_{cross}^s = \bigcup_{i \in [N], k \in [K]} \mathcal{E}_{cross}^{i,k} = \bigcup_{i \in [N], k \in [K]} \{(t, r_{t2p}, p_{i,k}), (p_{i,k}, r_{p2t}, t) | t \in \mathcal{T}_k^i\}. \tag{5}$$

Then, to augment classification, we connect the support NOI prompt node to the class node that its NOI belongs to, as illustrated by the few-shot section in Figure 2. That is, $\mathcal{E}_{supp} = \{(p_{i,k}, r_{s2c}, c_i) | i \in [N], k \in [K]\}$. Because the relation type $r_{s2c}$ differs from $r_{q2c}$ between the query NOI prompt node and class nodes, the model can differentiate information from query and support NOI. The overall components of the few-shot prompt graph $\mathcal{P}$ are

$$\mathcal{V}_p = \{p_q\} \bigcup \{p_{i,k} | i \in [N], k \in [K]\} \bigcup \{c_i | i \in [N]\},$$
$$\mathcal{E}_p = \mathcal{E}_{cross}^q \bigcup \mathcal{E}_{query} \bigcup \mathcal{E}_{cross}^s \bigcup \mathcal{E}_{supp}, \quad \mathcal{R}_p = \{r_{t2p}, r_{p2t}, r_{q2c}, r_{c2q}, r_{s2c}\}. \tag{6}$$

The prompted graph can be constructed in a similar way as discussed above. Like in the aforementioned scenarios, the output embeddings of class nodes are used to make binary classifications. OFA utilizes few-shot support examples by connecting the support NOI prompt nodes to the corresponding class nodes, and the model synthesizes both exemplary information and the task semantic information on the class nodes for more accurate predictions. Note that the few-shot class node representations are still consistent with that in zero-shot scenarios, so they can also be trained together.

To summarize, NOI represents the set of nodes related to the task, and the extracted NOI subgraph includes the neighborhood information of NOI. Then, the NOI prompt node summarizes information in the NOI by a graph learning model because all NOI nodes connect to the NOI prompt node. The NOI prompt node is later connected to a set of class nodes with text descriptions. After graph model processing, class node representations contain class information, task information, and NOI information, which can be used to make predictions independently, just like a prompted input to LLM contains the input, target, and task description. The implementation details and training procedure of the model can be found in Appendix C.

## 4 RELATED WORKS

The success of the LLM and prompt learning has enlightened many recent works that try to incorporate similar ideas to graphs. The first line of research tries to design prompt learning in the graph domain. Both VNT (Tan et al., 2023) and GraphPrompt (Liu et al., 2023c; Yu et al., 2023) introduce trainable prompt vectors to extract related information for different downstream tasks. HG-PROMPT (Yu et al., 2024a) further extends GraphPrompt to heterogeneous graphs. Prodigy (Huang et al., 2023) converts classification tasks to link prediction problems on prompt graphs, facilitating in-context graph learning. All-in-one (Sun et al., 2023) proposes to learn prompt graphs from downstream tasks. Our work also uses prompting to unify all classification tasks. More importantly, unlike existing work that still needs to train separate GNNs in different domains, we leverage language models to unify different tasks further and use one GNN to solve all tasks. The second line of research combines language models with GNNs. Some works directly apply LLMs to solve graph problems by describing the graph using natural language and feeding text description to LLMs, including InstructGLM (Ye et al., 2023), GraphText (Zhao et al., 2023b), NLGraph (Wang et al., 2023) and GPT4Graph (Guo et al., 2023). However, these methods also describe graph connections by texts, losing important structural features, while our method explicitly utilizes the information through GNNs. Very recently, GraphGPT (Tang et al., 2023) introduced a new method to encode the graph structure information with trainable vectors and an alignment module. More related works can be found in Appendix A.

## 5 EXPERIMENTS

The experiment section assesses OFA's potential to serve as a graph foundation model by answering the following questions: **Q1**: How does replacing the raw node/edge features with text features

from LLM affect GNN performance? **Q2**: Using text as features for all graphs, is a single OFA GNN versatile to tasks in all domains? **Q3**: What is the effect of different LLMs? **Q4**: Is the proposed graph prompting paradigm effective in in-context learning? By answering these questions, we validate the approach of using TAGs and OFA's ability to solve various tasks, demonstrating the strong potential of using OFA as a unified graph foundation model. More experiment and training details can be found in Appendix D.

Table 2: Results on supervised learning (first).

| Task type Metric | Cora Link AUC ↑ | Cora[1] Node Acc ↑ | PubMed Link AUC ↑ | PubMed[1] Node Acc ↑ | ogbn-arxiv[1] Node Acc ↑ | Wiki-CS Node Acc ↑ | HIV Graph AUC ↑ |
|---|---|---|---|---|---|---|---|
| GCN | 90.40±0.20 | 78.86±1.48 | 91.10±0.50 | 74.49±0.99 | 74.09±0.17 | 79.07±0.10 | 75.49±1.63 |
| GAT | 93.70±0.10 | **82.76±0.79** | 91.20±0.10 | 75.24±0.44 | 74.07±0.10 | **79.63±0.10** | 74.45±1.53 |
| OFA-ind-st | 91.87±1.03 | 75.61±0.87 | 98.50±0.06 | 73.87±0.88 | 75.79±0.11 | 77.72±0.65 | 73.42±1.14 |
| OFA-st | 94.04±0.49 | 75.90±1.26 | 98.21±0.02 | 75.54±0.05 | 75.54±0.11 | 78.34±0.35 | 78.02±0.17 |
| OFA-e5 | 92.83±0.38 | 72.20±3.24 | 98.45±0.05 | 77.91±1.44 | 75.88±0.17 | 73.02±1.06 | **78.29±1.48** |
| OFA-llama2-7b | 94.22±0.48 | 73.21±0.73 | **98.69±0.10** | 77.80±2.60 | 77.48±0.17 | 77.75±0.74 | 74.45±3.55 |
| OFA-llama2-13b | **94.53±0.51** | 74.76±1.22 | 98.59±0.10 | **78.25±0.71** | **77.51±0.17** | 77.65±0.22 | 76.71±1.19 |

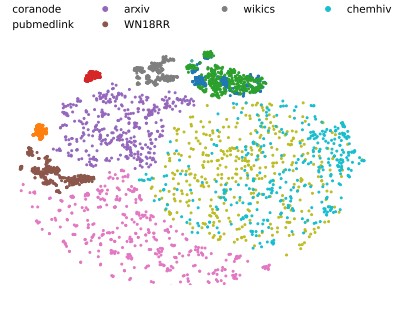

Table 3: Results on supervised learning (second).

| Task type Metric | WN18RR Link Acc ↑ | FB15K237 Link Acc ↑ | PCBA Graph APR ↑ |
|---|---|---|---|
| GCN | 67.40±2.40 | 74.20±1.10 | 20.20±0.24 |
| GIN | 57.30±3.40 | 70.70±1.80 | 22.66±0.28 |
| OFA-ind-st | 97.22±0.18 | **95.77±0.01** | 22.73±0.32 |
| OFA-st | 96.91±0.11 | 95.54±0.06 | 24.83±0.10 |
| OFA-e5 | 97.84±0.35 | 95.27±0.28 | **25.19±0.33** |
| OFA-llama2-7b | 98.08±0.16 | 95.56±0.05 | 21.35±0.94 |
| OFA-llama2-13b | **98.14±0.25** | 95.69±0.07 | 21.54±1.25 |

Figure 3: Output embedding space of NOI prompt nodes on all datasets for OFA-joint-st.

## 5.1 CROSS-DOMAIN SUPERVISED LEARNING

To answer **Q1-Q3**, we conduct experiments on the supervised learning scenario using all collected OFA datasets with different LLMs. Specifically, we select four popular LLMs for evaluation, including sentence transformer (Reimers & Gurevych, 2019), e5-large-v2 (Wang et al., 2022a), Llama2-7b, and Llama2-13b (Touvron et al., 2023). Then, we evaluate OFA in two different settings. The first setting trains and tests the model on each dataset independently with text embedding generated by the sentence transformer, denoted as OFA-ind-st. The second setting trains a single model using all datasets jointly. We denote the joint model utilized different LLMs as OFA-st, OFA-e5, OFA-llama2-7b, and OFA-llama2-13b respectively. For baseline methods, we use GCN (Kipf & Welling, 2017), GAT (Veličković et al., 2018), and GIN (Xu et al., 2018) for fair comparison. The results can be found in Table 2 and Table 3.

From the results, we have the following observations: (1) Both the independent and joint training achieve comparable or better results on all datasets compared to baseline methods. (2) OFA successfully enabled a single graph model to be effective on all graph datasets across different domains as the joint version with all different LLMs performs well on all datasets. Further, we can see that the joint version OFA-st achieves better results on most of the datasets compared to OFA-ind-st. This may indicate that by leveraging the text feature and GPP, the knowledge learned from one domain/dataset can be useful for the learning of other domains/datasets. (3) The comparison of different LLMs is interesting, generally speaking, a larger LLM can achieve better and more stable performance in joint training. We also observe a faster convergence for larger LLM (Llama2-13b). However, the margin is less significant. Meanwhile, different LLMs seem specialized for different domains. For example, Llama2 achieves better performance in citation networks but e5-large-v2 achieves great results in molecular datasets. To further investigate the mechanism behind the OFA,

---

[1]The split we use differs from original one, see Apeendix D for details.

we take the output embedding of NOI prompt nodes from OFA-joint-st for each dataset and project it to two-dimensional space. As shown in Figure 3, node embeddings from different domains are separated. This demonstrates that the OFA model can represent data from different domains in different sub-spaces to process it. In Appendix E, we conduct additional ablation studies to verify the effectiveness of the proposed GPP.

## 5.2 FEW-SHOT AND ZERO-SHOT LEARNING

To answer **Q4**, we design few-shot and zero-shot scenarios for all levels of tasks. We consider both transductive and transfer situations. In the transductive setting, the task is to classify unseen classes on the same training graph. In the transfer setting, both the test graph and test classes are unseen. For simplicity, all experiments are conducted using OFA datasets generated from the sentence transformer. We train one few-shot model on various tasks varying $N$-way and $k$-shot, where $N \geq 2$ and $k \geq 0$. We include ogbn-arxiv, FB15K237, and Chemble as training sets, then evaluate two transductive settings: ogbn-arxiv and FB15K237, and four transfer settings: Cora, WN18RR, HIV, and PCBA. We present node/link/graph level results in Table 4 / 5 / 6, respectively. The model is denoted by OFA-joint-lr (**l**ow-**r**esource).

Table 4: Few-shot and Zero-shot results (Acc) on ogbn-arxiv and Cora (Node-level).

| # Way | ogbn-arxiv-5-way (Transductive) | | | | Cora-2-way (Transfer) | | |
|---|---|---|---|---|---|---|---|
| Task | 5-shot | 3-shot | 1-shot | 0-shot | 5-shot | 1-shot | 0-shot |
| GPN | $50.53_{\pm 3.07}$ | $48.32_{\pm 3.80}$ | $38.58_{\pm 1.61}$ | - | $63.83_{\pm 2.86}$ | $56.09_{\pm 2.08}$ | - |
| TENT | $60.83_{\pm 7.45}$ | $56.03_{\pm 8.90}$ | $45.62_{\pm 10.70}$ | - | $58.97_{\pm 2.40}$ | $54.33_{\pm 2.10}$ | - |
| GLITTER | $56.00_{\pm 4.40}$ | $57.44_{\pm 4.90}$ | $47.12_{\pm 2.73}$ | - | - | - | - |
| TLP-BGRL | $50.13_{\pm 8.78}$ | $46.21_{\pm 7.92}$ | $35.81_{\pm 8.58}$ | - | $81.31_{\pm 1.89}$ | $59.16_{\pm 2.48}$ | - |
| TLP-SURGL | $77.89_{\pm 6.46}$ | $74.19_{\pm 7.55}$ | $61.75_{\pm 10.07}$ | - | $92.49_{\pm 1.02}$ | $81.52_{\pm 2.09}$ | - |
| Prodigy | $61.09_{\pm 5.85}$ | $58.64_{\pm 5.84}$ | $48.23_{\pm 6.18}$ | - | - | - | - |
| OFA-joint-lr | $61.45_{\pm 2.56}$ | $59.78_{\pm 2.51}$ | $50.20_{\pm 4.27}$ | $46.19_{\pm 3.83}$ | $76.10_{\pm 4.41}$ | $67.44_{\pm 4.47}$ | $56.92_{\pm 3.09}$ |

For node-level tasks, we compare with meta-learning models (Ding et al., 2020; Wang et al., 2022c;b) and graph contrastive learning model (Tan et al., 2022). For graph-level tasks, LLM-based models Galactica-1.3B (Taylor et al., 2022) and GIMLET (Zhao et al., 2023a) are considered. Prodigy Huang et al. (2023) follows a different setting, where the model is trained on the MAG240M or Wiki datasets (Hu et al., 2021) and transferred to the corresponding tasks. OFA exhibits comparable or better performance than most existing works on few-shot tasks. Especially in the transfer setting of node tasks, where all baselines are trained and evaluated on the Cora dataset, OFA still shows comparable performance without any prior knowledge about the test dataset, illustrating its capability to generalize. Furthermore, our proposed GPP endows OFA with the ability to address zero-shot scenarios—a task generally impossible for most existing baseline models. OFA utilizes one single model to address all low-resource tasks across domains, demonstrating the ability of in-context learning.

Table 5: Few-shot and Zero-shot results (Acc) on FB15K237 and WN18RR (Link-level).

| # Way | FB15K237-20-way (Transductive) | | WN18RR-5-way (Transfer) | |
|---|---|---|---|---|
| Task | 5-shot | 0-shot | 5-shot | 0-shot |
| Prodigy | $74.92_{\pm 6.03}$ | - | - | - |
| OFA-joint-lr | $82.56_{\pm 1.58}$ | $70.20_{\pm 2.40}$ | $46.32_{\pm 4.18}$ | $30.96_{\pm 5.46}$ |

Table 6: Few-shot and Zero-shot results (AUC) on HIV and PCBA (Graph-level & Transfer setting).

| # Way | HIV-2-way | | PCBA-2-way | |
|---|---|---|---|---|
| Task | 5-shot | 0-shot | 5-shot | 0-shot |
| Galactica-1.3B | - | 33.85 | - | 52.02 |
| GIMLET | - | 66.24 | - | 62.11 |
| OFA-joint-lr | $63.58_{\pm 1.81}$ | $35.67_{\pm 4.46}$ | $51.53_{\pm 9.94}$ | $60.62_{\pm 5.45}$ |

## 6 CONCLUSIONS, LIMITATIONS AND FUTURE RESEARCH

In this work, we propose OFA, the first solution towards building the foundation GNN model for learning on graphs. By showing great results on supervised, few-shot, and zero-shot scenarios, OFA reveals great potential as the future foundation model on the graph. Currently, OFA falls short of learning regression tasks and the cross-domain datasets are limited, we leave this to future work. More discussion can be found in Appendix F.

## ACKNOWLEDGEMENT

Hao Liu, Jiarui Feng, Lecheng Kong, and Yixin Chen are supported by NSF grant CBE-2225809. Muhan Zhang is supported by the National Key R&D Program of China (2022ZD0160303) and National Natural Science Foundation of China (62276003).

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

## A    RELATED WORKS (EXTENDED)

**Large Language Model and Prompt Learning.**    Since the ground-breaking advances in Large Language Models, including GPT (Brown et al., 2020; Wei et al., 2022) and LLaMa (Touvron et al., 2023), tremendous attention has been drawn to developing and using them. One particular approach is prompt learning. By prompting the LLMs according to a carefully designed paradigm, users can uncover LLM's surprising capability in many difficult tasks. Chain-of-thoughts, by providing step-by-step reasoning examples, greatly improve LLMs' reasoning power. Considerable efforts were also made to use prior semantic knowledge learned by LLMs to perform zero-shot and few-shot tasks (Reynolds & McDonell, 2021; Liu et al., 2023b; Chowdhery et al., 2022). The key advantage of prompting LLMs is that no further fine-tuning is required for the model to adapt to a new class of tasks. Innovated by this practical advantage, several works adapted the prompting idea to the GNN domain. VNT (Tan et al., 2023) offers a new method for few-shot node classification (FSNC) by integrating trainable virtual node embeddings into the original graph, which serves as a form of prompting. For each new FSNC task, the pre-trained graph transformer model can be frozen and only the embedding of virtual nodes needs to be trained. GraphPrompt (Liu et al., 2023c; Yu et al., 2023) also introduces trainable prompt vectors to extract related information for the downstream tasks. Meanwhile, it further utilizes the subgraph and link prediction to design a novel unsupervised pertaining method, which unifies the pertaining and downstream tasks. MultiGPrompt Yu et al. (2024b) proposes a multi-task pre-training and prompting framework, effectively harnessing diverse pretext task knowledge for enhanced graph-based analysis. Prodigy Huang et al. (2023) converts classification tasks to link prediction problems on prompt graphs, facilitating in-context graph learning. All-in-one Sun et al. (2023) proposes to learn prompt graphs from data. Our work also uses prompting to unify all classification tasks. However, our works differ in two parts. First, unlike existing work that still needs to train separate GNNs in different domains, we leverage language models to unify different tasks further and use one GNN to solve all tasks. Second, most of the existing works cannot perform zero-shot learning as their prompt module needs to be re-trained for different downstream tasks. instead, OFA directly describes all tasks in a unified way, which allows the model to perform zero-shot learning without further training.

**Graph Neural Networks.** Recently, extensive efforts have been spent on using GNNs to learn relational data. Earlier GNN variants, including GCN (Kipf & Welling, 2017), GAT (Veličković et al., 2018), and GraphSage (Hamilton et al., 2017), achieved great success in solving graph learning problems. Later works unify the GNNs into the Message Passing Neural Network (Gilmer et al., 2017) and show the expressivity upper-bound of such framework (Xu et al., 2018; Morris et al., 2019), which demonstrates the inherent difference between learning on graphs and learning on other data formats such as images and languages where expressivity is not an issue. Following such observation, subsequent works, such as SEAL (Zhang & Chen, 2018) and ID-GNN (You et al., 2021), propose subgraph GNNs that apply GNNs to subgraphs and aggregate subgraph representations to enhance the expressivity. Moreover, unlike MPNN, which fits the full graph into the GPU memory, subgraph GNNs process the data by subgraphs, solving the scalability problem of MPNN. Innovated by subgraph GNNs, our work and concurrent works (Huang et al., 2023; Sun et al., 2023) also add prompts to subgraphs of interest, which allows our models to perform cross-level tasks.

The applications of GNN have also become prevalent. Much research focuses on the molecular property prediction domain and has achieved promising performance (Zhang et al., 2023a; Feng et al., 2023; Kong et al., 2023; Feng et al., 2022). GNNs have also become one of the primary tools in citation network analysis and mining (Kipf & Welling, 2017; Chien et al., 2022). Many efforts have also been made to adapt GNNs to the Knowledge Graph mining domain and achieved great success due to GNN's efficiency and inductive learning capabilities (Zhu et al., 2021; Kong et al., 2022). While the promising results of these applications demonstrate the potential of GNNs, a critical problem is that they are tailored to the specified application. On the contrary, our work, by three carefully designed components, unifies all tasks into one GNN framework that allows large-scale training and inference across different domains. The proposed framework can be a foundation model in the graph learning domain.

**Language Models and Graph Neural Networks.** The surge of foundational Large Language Models inspires several directions combining language models with GNNs. One of them directly applies LLMs to solve graph problems. They propose graph descriptive languages to represent the structural information as prompt texts and feed them to LLMs, like InstructGLM (Ye et al., 2023), Graph-

Text (Zhao et al., 2023b), NLGraph (Wang et al., 2023), and GPT4Graph (Guo et al., 2023). Some works for molecular graph classification propose to only use SMILE molecule sequences to describe the molecular graph in input to the LLM, including LMMOL (Qian et al., 2023) and GIMLET (Zhao et al., 2023a). The exceptional power of LLMs enables these methods to perform difficult tasks such as zero/few-shot learning. However, such graph representation is implicit, and the LLM might not correctly capture the structural information but only summarize texts corresponding to the nodes that appear in the prompt. Such approaches will fall short in applications where graph structures are important.

Very recently, GraphGPT (Tang et al., 2023) introduced a new method to encode the graph structure information with trainable vectors and an alignment module. It concatenates the trainable vectors along with textual input to LLms to improve the structure reasoning ability of LLMs. Another direction uses LLMs to encode the corresponding texts of nodes, such as paper abstracts, and apply GNN to the graph with encoded text embeddings (He et al., 2023; Chen et al., 2023). The high-quality text representation from LLMs allows these models to achieve better performance. Our approach adopts the same idea of using text embedding to represent graph entities (nodes and edges). However, we are the first work that drives the language model to full power by systematically unifying graph entities in different domains with the same language protocol and representing them in the same embedding space. This consequently grants cross-domain functionality that other GNN frameworks lack. Table 7 summarizes the characteristics of different models. We can see that OFA is the most versatile model among existing works.

Table 7: A comparison between OFA and related methods.

| | In-context | Few-shot | Zero-shot | Cross-tasks | Cross-domain | GNN |
|---|---|---|---|---|---|---|
| LMMOL | ✓ | ✓ | ✓ | | | |
| GIMLET | ✓ | ✓ | ✓ | | | |
| InstructGLM | ✓ | ✓ | ✓ | ✓ | ✓ | |
| GraphText | ✓ | ✓ | ✓ | ✓ | ✓ | |
| NLGraph | ✓ | ✓ | ✓ | ✓ | ✓ | |
| GPT4GRAPH | ✓ | ✓ | ✓ | ✓ | ✓ | |
| ExpAsFeat | ✓ | | | | | ✓ |
| VNT | ✓ | ✓ | | | | ✓ |
| LLMForGraph | ✓ | ✓ | ✓ | | | ✓ |
| PRODIGY | ✓ | ✓ | | | | ✓ |
| GraphPrompt | ✓ | ✓ | | ✓ | | ✓ |
| All-in-One | ✓ | ✓ | | ✓ | | ✓ |
| OFA | ✓ | ✓ | ✓ | ✓ | ✓ | ✓ |

# B  MORE ON OFA DATASET

## B.1  DATASET COLLECTION AND CONSTRUCTION

In this section, we discuss the detailed processing procedure for each dataset collected in OFA.

### B.1.1  CORA

Cora is a citation network that contains papers and their citation relationship in the computer science domain. The raw text data of the Cora dataset was collected from the GitHub repository provided in Chen et al. (2023). Each node in Cora represents a research paper from the computer science domain. The raw text feature of a node is the title and abstract of the respective paper. Every edge in the Cora dataset indicates the citation relationship between papers. Each node's label corresponds to the category of the paper. Tasks that can be executed on Cora include predicting the paper's category (node-level) or identifying missing citation links within the graph (link-level). Using the proposed processing protocol, we reformat all text features in Cora. Particularly, for each node category, we further use gpt-3.5-turbo(ChatGPT) to generate a description as additional information. In table 8, we show a processed example on Cora dataset.

### B.1.2  PUBMED

Cora is a citation network that contains papers and their citation relationship in the biomedical domain. The raw text data of the PubMed dataset was collected from the GitHub repository provided in Chen et al. (2023). All nodes and edges are similar to Cora dataset and we use the exact same processing procedure as Cora dataset. Because its original literature categories are diabetes, experimental/diabetes, type 1/diabetes, and type 2, which are overly simple and very difficult for our BERT-based LLM to distinguish, we asked ChatGPT to generate a detailed description of each category.

Table 8: A processed example on Cora/Pubmed/ogbn-arxiv dataset.

| Node type | Text feature |
|---|---|
| Input graph node | feature node. literature category and description: <*title*>. <*abstract*> |
| Input graph edge (Cora/Pubmed) | feature edge. co-citation |
| Input graph edge (OGBN-ARXIV) | feature edge. citation |
| Node classification task | |
| Class node | prompt node. literature category and description: <*category name*>. <*description*> |
| Prompt node | prompt node. node classification of literature category. |
| Link prediction task | |
| Class node | prompt node. two papers (do not have/have) co-citation |
| Prompt node | prompt node. link prediction on the papers that are cited together |

Table 9: A processed example on Wiki-CS dataset.

| Node type | Text feature |
|---|---|
| Input graph node | feature node. wikipedia entry name: <*entry name*>. Entry content: <*entry content*> |
| Input graph edge | feature edge. wikipedia page link. |
| Node classification task | |
| Class node | prompt node. wikipedia entry category: <*category name*> |
| Prompt node | prompt node. node classification of wikipedia entry category. |

### B.1.3 OGBN-ARXIV

Cora is a citation network that contains papers and their citation relationship collected from Arxiv platform. The raw text data of the ogbn-arxiv was collected using the same protocol as the GitHub repository provided in Prodigy (Huang et al., 2023). For ogbn-arxiv, we only evaluate the model on the node classification task. All nodes and edges are similar to Cora dataset and we use the exact same processing procedure as Cora dataset except that the description for each category is directly obtained from the Prodigy.

### B.1.4 WIKI-CS

Wiki-CS is an Internet link network with each node representing a Wikipedia page and each edge representing the reference link. The raw text of the Wiki-CS dataset was collected from the official website (Mernyei & Cangea, 2020). The raw text feature of a node is the name and content of an entry in Wikipedia. Each node's label corresponds to the category of the entry. We evaluate Wiki-CS on node classification tasks. In table 9, we show a processed example on the Wiki-CS dataset using the processing protocol proposed in OFA.

### B.1.5 FB15K237

FB15K237 is a knowledge graph that contains knowledge base relation triples and textual mentions of Freebase entity pairs. The raw text data of nodes in FB15K237 was collected from GitHub repository[2]. The raw text feature of a node is the name of the relation entity and its description. The

---

[2]https://github.com/villmow/datasets_knowledge_embedding/tree/master

Table 10: A processed example on FB15K237 dataset.

| Node type | Text feature |
|---|---|
| Input graph node | feature node. entity and entity description: *\<entity name\>. \<entity description\>* |
| Input graph edge | feature edge. relation between two entities: *\<relation name\>* |
| Link prediction task | |
| Class node | prompt node. relation between two entities: *\< relation name \>* |
| Prompt node | prompt node. relation type prediction between the connected entities. |

Table 11: A processed example on Molecule datasets.

| Node type | Text feature |
|---|---|
| Input graph node | feature node. atom. \<element name\>, \<atom chirality\>, degree of \<atom degree\>, formal charge of \<formal charge\>, num of hydrogen is \<number of hydrogen\>, num of radical electron is \<number of radical electrons\>, hybridization is \<hybridization\>, (is/is not) aromatic, (is/is not) in ring. |
| Input graph edge | feature edge. chemical bond. \<bond type\> bond, bond stereo is \<bond stereo\>, (is/is not) conjugated |
| Node classification task | |
| Class node | prompt node. molecule property description. \<molecule property description. e.g. the molecule is effective to the following assay:...\> |
| Prompt node | prompt node. Graph classification on molecule properties. |

raw text feature of an edge is the type of relation between two entities. We provide an example of processed data example in Table 10.

### B.1.6 WN18RR

WN18RR is a knowledge graph, which is a subset of WordNet that consists of 18 relations and 40943 entities. The raw text data of nodes in WN18RR was collected from GitHub repository[2]. The raw text feature of nodes and edges are the same as FB15K237 and we follow the same process protocol to process the WN18RR dataset.

### B.1.7 MOLECULAR DATASETS

We adopted three molecular datasets: (1) ChEMBL dataset Gaulton et al. (2012) is a widely used molecule property prediction dataset. It contains 1,310 prediction target labels of molecules from biological assays for drug discovery. (2) MOLPCBA dataset Wu et al. (2018) is a subset of the BioChem BioAssay dataset consisting of 128 labels on the biological activities of small molecules. (3) MOLHIV dataset Wu et al. (2018) contains over 40,000 compounds labeled for their ability to inhibit HIV replication. The raw data of these datasets are represented in SMILE string format. We use RDKit to construct molecule objects and graph structures from the SMILE string and use natural language to describe the atoms (nodes) and bonds (edges) as shown in Table 11. For the class nodes text, we use the assay/molecule properties description in Zhao et al. (2023a).

### B.2 VISUALIZATION OF OFA DATASET

In this section, we provide a visualization of all generated OFA datasets using the sentence transformer. Concretely, For each generated dataset, we randomly select 400 node embeddings and project it to 2 dimensions using TSNE. Note for molecular datasets, we include all node embeddings. The result is shown in Figure 4.

We can see that the generated embeddings from different domains are successfully separated by the LLM as the embedding from molecular datasets, knowledge graphs, wiki pages, and citation networks are well separated in the visualization. Moreover, the LLM can even separate the citation network from different research domains. embeddings from Arxiv and Cora, which mainly contain papers from computer science are close to each other and far from the embeddings of Pubmed, which focus on biology. This result reveals one of the key rationales behind the success of the OFA. That is, by encoding the datasets from different domains into different sub-spaces, OFA allows one GNN to learn the information from different domains separately without impacting each other.

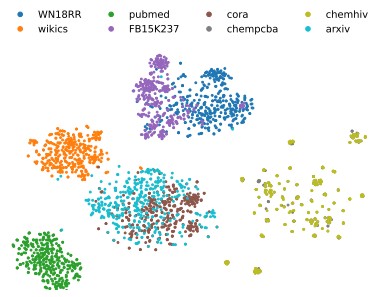

Figure 4: Embedded node features from all OFA datasets (sentence transformer).

## C  IMPLEMENTATION OF OFA

The last section described how OFA combines LLMs, text-attributed graphs, and the graph-prompting paradigm to facilitate cross-domain and in-context learning. In this section, we illustrate the training and evaluation procedure of OFA.

Training an OFA pipeline for different tasks is straightforward with the unified task representation introduced in Section 3 because all the inputs are standardized to the prompted text attributed graph $\mathcal{G}_m$ regardless of the task type. The task information is carried out in the prompt nodes and edges. Recall that OFA first embeds all texts in graphs to vector representations using LLM:

$$x_{ij} = LLM(s_{e_{ij}}), \forall e_{ij} \in \mathcal{E}_m, \quad x_i = LLM(s_{v_i}), \forall v_i \in \mathcal{V}_m. \tag{7}$$

Then, a GNN processes $\mathcal{G}_m$ along with the embedded texts through multiple message-passing layers and gets the final node embedding for each node in $\mathcal{G}_m$. Formally,

$$h_{v_i}^{l+1} = \boldsymbol{W}_{\text{self}}h_{v_i}^l + \sum_{r \in \mathcal{R}_m} \sum_{v_j \in \mathcal{N}_1^r(v_i)} \frac{1}{|\mathcal{N}_1^r(v_i)|} \boldsymbol{W}_r(\text{ReLU}(h_{v_j}^l + x_{ij})), \tag{8}$$

where $h_{v_i}^0 = x_i$, $\boldsymbol{W}_{\text{self}}$ and $\boldsymbol{W}_r$ are trainable transformation matrix for self-loop and relations, $\mathcal{N}_1^r(v_i)$ is the direct neighbors that connects to $v_i$ with relation type $r$. Our GNN model is a natural extension of R-GCN (Schlichtkrull et al., 2018), incorporating edge features. Such an implementation helps differentiate nodes in the input graph and the prompt graph. It also effectively extracts useful information from the input graph and encodes it to the class node in the prompt graph for prediction. Usually, the last layer output can be used for prediction. However, due to the different natures of tasks and the well-known over-smoothing effect, a GNN with a fixed number of layers will not perform well on all tasks. Hence, we design a simple attention layer to summarize all layer outputs from $h^1$ to $h^L$. Eventually, the node representation is

$$h_{v_i} = \boldsymbol{H} \cdot Softmax((\boldsymbol{W}_k\boldsymbol{H})' \cdot \boldsymbol{W}_q x_i)', \quad \boldsymbol{H} = [h_{v_i}^1 \quad ... \quad h_{v_i}^l \quad ... \quad h_{v_i}^L], \tag{9}$$

where $\boldsymbol{W}_k$ and $\boldsymbol{W}_q$ are key and query trainable weights. Note that because the text features $x_i$ also include the domain information, by querying on the feature, a properly trained attention layer can choose the most important layers to aggregate outputs based on the domain. Lastly, we gather the output node embedding for each class node $\{h_{c_i}|\forall i \in [N]\}$ and use an MLP prediction head to perform binary classification on every class node. Even for multi-class problems, each class node connects to the NOI prompt node and hence can integrate the information of other connected class nodes through message-passing. While each class node is processed separately by the MLP, such binary classification is still conditioned on other candidate classes and not individualized. Moreover, since the format of $\mathcal{G}_m$ is consistent in different learning scenarios (supervised/few-shot/zero-shot) and task types (node/link/graph-level), such a procedure can work without any modification across these situations. Formally, the likelihood of class $i$ is:

$$p_i = \sigma(\text{MLP}(h_{c_i})), \tag{10}$$

where $\sigma$ is the Sigmoid function. If it's a multi-class problem, OFA collects all classes' probabilities as

$$l_p = \text{argmax}\left((p_i|i \in [N])\right), \tag{11}$$

and $l_p$ is the final prediction from the model. The binary training loss for a single $\mathcal{G}_m$ can be expressed as:

$$\mathcal{L}_{\mathcal{G}_m} = -\frac{1}{N}\sum_{i=1}^{N} y_i \cdot log(p_i) + (1 - y_i) \cdot (1 - log(p_i)), \tag{12}$$

where $N$ is the number of candidate classes in $\mathcal{G}_m$.

## D EXPERIMENTAL SETTINGS

In this section, we provide detailed settings for all experiments shown in the paper. The code of OFA can be found at the Github link `https://github.com/LechengKong/OneForAll`.

### D.1 SUPERVISED LEARNING EXPERIMENTS

We set the number of layers for GNN of independent training (OFA-ind-st) and joint training to be 6 and 7 respectively, as the joint model might require a larger embedding space. The dropout rate is set to 0.15. We set the hidden size of the GNN as 768 and project the initial embedding generated by different LLMs to 768 before the GNN. We train OFA-ind-st for 100 epochs and 50 for OFA-st and OFA-e5. For OFA-llama2-7b and OFA-llama2-13b, we only train 40 epochs due to the limitation of computing resources. For OFA-ind-st, we evaluate the model after each epoch and take the model with the best validation performance for testing. For all versions of joint training, since there is no easy way to define the best validation performance, we directly use the final model for testing. Because datasets vary in sample size, smaller sets might be neglected during training. Hence, we sample from each dataset by a multiple of the dataset size to form the training set and control the weight of each dataset. Data are resampled after every epoch. The multipliers are Cora-link: 1.5, Cora-node 2, Pubmed-link: 0.5, Pubmed-node: 2.5, ogbn-arxiv: 0.7, WN18RR: 0.6, FB15K237: 0.4, WikiCS: 2, ChEMBL: 1, PCBA: 2, HIV: 4. For independent and joint training, we repeat experiments 10 and 3 times, respectively, and report the mean results and standard deviation. The final training set contains all training sets of the collected datasets, and the test sets are the datasets' original test sets.

For Cora-node, Pubmed-node, and ogbn-arxiv datasets, a different split is used. Specifically, for Cora-node and Pubmed-node, the split is obtained from Chen et al. (2023), where they split data into 10 folds and we use the first fold as our split. For ogbn-arxiv, in each experiment, we will randomly split data with a train/val/test ratio of 0.8/0.1/0.1. For Cora-node, Pubmed-node, ogbn-arxiv, WN18RR, and FB15K237, we rerun the baseline models 10 times and report the average performance. For other datasets, we report results from existing works.

### D.2 FEW-SHOT AND ZERO-SHOT LEARNING EXPERIMENTS

For OFA joint training low resource (OFA-joint-lr) experiments, we set the number of layers to be 5, the learning rate as 0.0001, and the number of epochs to be 30. We split the label of ogbn-arxiv dataset with ratio $[20, 10, 10]$ for train/validation/test, and use the $[142, 47, 48]$ as the split ratio for FB15K237 dataset. During training, we use the training set from Arxiv (node-level), the training set from FB15K237 (link-level), and the whole Chemble dataset (graph-level) to train a single model for all downstream tasks. We construct diverse $N$-way $k$-shot tasks from the training sets. For ogbn-arxiv dataset, $N$ varies from 3 to 5, $k$ varies from 0 to 5; for FB15K237 dataset, $N$ varies from 3 to 10, $k$ varies from 0 to 5; for ChEMBL dataset, all tasks are 2-way, and $k$ varies from 0 to 10.

In few-shot graph construction, the text feature of class nodes differs significantly from that in supervised and zero-shot scenarios. Specifically, we omit category information and utilize a uniform text feature across all class nodes. This uniformity shifts the focus from learning class-specific information to enhancing the model's ability to compare the query node with support nodes. Such comparisons are crucial for determining matches, which proves particularly effective in few-shot scenarios where the model must classify unseen classes.

During the test, we involved six datasets: for the transductive setting, we evaluate on the test set from ogbn-arxiv dataset and test set from FB15K237 dataset; for the transfer setting, we evaluate on Cora (node-level), WN18RR (link-level), MOLPCBA and MOLHIV (graph-level). Note that for

the transductive setting, we keep the same train/validation/test labels for OFA and all the baselines to ensure a fair comparison. For baseline results on the Cora dataset, we report the performance provided by COLA (Liu et al., 2023a), which is the average performance of 20 random label splits. For baseline results of Prodigy, we follow the provided code to pre-train separate models for different shot numbers, that is, we pre-train 5-shot/3-shot/1-shot models for node-level tasks and pre-train 5-shot/3-shot/1-shot models for link-level tasks, then evaluate different shot settings using the corresponding pre-trained model. For the baselines of graph-level tasks, we report the results from GIMLET (Zhao et al., 2023a).

Table 12: Ablation study on different prompting design.

| | Joint | | Separate | |
|---|---|---|---|---|
| | Full | - Class node | Full | - Class node |
| ogbn-arxiv | $75.23\pm0.03$ | $75.19\pm0.10$ | $75.06\pm0.08$ | $75.39\pm0.09$ |
| HIV | $75.92\pm0.45$ | $71.60\pm0.54$ | $75.81\pm0.36$ | $75.43\pm0.50$ |
| Cora-node | $75.48\pm0.29$ | $71.39\pm0.07$ | $75.72\pm0.49$ | $76.12\pm0.87$ |
| Cora-link | $92.27\pm0.84$ | $89.51\pm0.46$ | $93.16\pm0.95$ | $92.80\pm0.62$ |

Table 13: Few-shot results (Acc) on ogbn-arxiv (Node-level).

| # Way | 5-way | | | | 3-way | | | |
|---|---|---|---|---|---|---|---|---|
| Task | 5-shot | 3-shot | 1-shot | 0-shot | 5-shot | 3-shot | 1-shot | 0-shot |
| GPN | $50.53\pm3.07$ | $48.32\pm3.80$ | $38.58\pm1.61$ | - | $62.25\pm4.94$ | $58.52\pm3.00$ | $48.45\pm5.60$ | - |
| TENT | $60.83\pm7.45$ | $56.03\pm8.90$ | $45.62\pm10.70$ | - | $74.20\pm9.93$ | $70.48\pm11.50$ | $59.38\pm13.55$ | - |
| GLITTER | $56.00\pm4.40$ | $57.44\pm4.90$ | $47.12\pm2.73$ | - | $62.13\pm10.85$ | $60.93\pm12.12$ | $59.20\pm5.48$ | - |
| BGRL | $50.13\pm8.78$ | $46.21\pm7.92$ | $35.81\pm8.58$ | - | $62.93\pm11.74$ | $58.37\pm11.34$ | $46.30\pm10.83$ | - |
| SURGL | $77.89\pm6.46$ | $74.19\pm7.55$ | $61.75\pm10.07$ | - | $86.27\pm7.54$ | $83.75\pm8.86$ | $73.46\pm12.68$ | - |
| Prodigy | $61.09\pm5.85$ | $58.64\pm5.84$ | $48.23\pm6.18$ | - | $73.64\pm6.93$ | $71.43\pm7.28$ | $61.59\pm8.53$ | - |
| OFA-joint-lr | $61.45\pm2.56$ | $59.78\pm2.51$ | $50.20\pm4.27$ | $46.19\pm3.83$ | $73.22\pm2.65$ | $72.24\pm3.81$ | $60.60\pm3.71$ | $58.87\pm3.37$ |
| OFA-ind-lr | $59.92\pm1.32$ | $58.68\pm6.40$ | $52.80\pm3.94$ | $46.56\pm0.82$ | $72.18\pm3.33$ | $71.80\pm1.59$ | $60.47\pm2.65$ | $64.13\pm0.98$ |

Table 14: Few-shot results (Acc) on FB15K237 (Link-level).

| # Way | 20-way | | | | 10-way | | | |
|---|---|---|---|---|---|---|---|---|
| Task | 5-shot | 3-shot | 1-shot | 0-shot | 5-shot | 3-shot | 1-shot | 0-shot |
| Prodigy | $74.92\pm6.03$ | $70.32\pm6.30$ | $55.49\pm6.88$ | $5.11\pm3.07$ | $84.30\pm7.80$ | $79.61\pm8.28$ | $66.10\pm9.89$ | $10.02\pm6.46$ |
| OFA-joint-lr | $82.56\pm1.58$ | $81.33\pm2.54$ | $75.39\pm2.86$ | $70.20\pm2.40$ | $90.44\pm1.95$ | $89.68\pm1.38$ | $86.38\pm1.60$ | $70.84\pm2.31$ |
| OFA-ind-lr | $87.43\pm1.56$ | $86.42\pm2.45$ | $83.87\pm2.88$ | $72.24\pm3.45$ | $93.44\pm1.57$ | $92.44\pm0.84$ | $89.16\pm1.11$ | $82.08\pm0.81$ |

Table 15: Few-shot results (Acc) on WN18RR (Link-level).

| # Way | 10-way | | | | 5-way | | | |
|---|---|---|---|---|---|---|---|---|
| Task | 5-shot | 3-shot | 1-shot | 0-shot | 5-shot | 3-shot | 1-shot | 0-shot |
| OFA-joint-lr | $31.42\pm1.74$ | $28.46\pm1.83$ | $26.24\pm1.96$ | $19.98\pm3.26$ | $46.32\pm4.18$ | $44.20\pm4.58$ | $33.86\pm3.41$ | $30.96\pm5.46$ |
| OFA-ind-lr | $32.64\pm1.56$ | $30.56\pm1.02$ | $25.82\pm1.07$ | $20.40\pm2.86$ | $48.32\pm3.19$ | $45.04\pm2.39$ | $34.40\pm1.47$ | $38.24\pm1.76$ |

# E MORE EXPERIMENTAL RESULTS

This section includes more experimental results that provide a detailed evaluation of OFA's in-context learning ability and justify the OFA design by ablation study.

### E.1 Ablation study

While using LLM to unify graph and task representation is a unique design in OFA, there are alternatives to our prompting paradigm GPP to make classifications. One alternative is to discard graph prompting completely but keep track of the NOI for each NOI subgraph. After being processed by the graph model, the embeddings of the NOI are summarized by average pooling to form the final NOI representation, the LLM encoded tasks representation is concatenated to the NOI for binary classification. We denote this as **"-Class node"**. We perform a case study comparing the two methods on hiv, ogbn-arxiv, Cora-node, and Cora-link datasets when these datasets are trained jointly using the same model and separately. The results are presented in Table 12.

We observe that, if the datasets are trained separately, all methods achieve similar performance, since in end-to-end training for one dataset the prompting design is essentially a pooling mechanism. However, it is striking that the **"-Class node"** approach's performance significantly drops when the datasets are trained together, while the OFA prompting approach maintains the original performance. Intuitively, when datasets from different domains are trained together, the model needs to learn which domain a particular data is from and which tasks are being performed to make appropriate predictions. However, without the NOI prompt node that carries task text descriptions, the **"-Class node"** approaches can confound tasks in different domains.

### E.2 Few-shot and Zero-shot Results

To explore the in-context learning ability of OFA, we train a single model for all few-shot and zero-shot low-resource tasks denoted as OFA-joint-lr. Here, we provide more comprehensive experiment results spanning more ways and shots. We also implement three experiments that train separately on node-, link-, and graph-level tasks denoted as OFA-ind-lr. For node-level tasks, we train the model using the ogbn-arxiv dataset. For link-level tasks, we train the model on the FB15K237 dataset. For graph-level tasks, we train the model on Chemble.

Results for the ogbn-arxiv and Cora can be found in Table 13 and Table 16, respectively. We can see that for the Cora dataset, OFA-joint-lr achieve better performance than OFA-ind-lr in all setting. This may indicate that the knowledge learned from other task levels like link- or graph-level can help the generalization of the model on Cora. For ogbn-arxiv dataset, the results for OFA-joint-lr and OFA-ind-lr are similar.

The results for FB15K237 and WN18RR datasets can be found in Table 14 and Table 15, respectively. For link-level task, the OFA-ind-lr have better results than OFA-joint-lr. This may indicate that the knowledge learned from graph or node level is not helpful for link-level tasks.

Finally, the results for HIV and PCBA datasets can be found in Table 17 and Table 18, respectively. We can notice that joint training can benefit the prediction of HIV in most cases and the zero-shot scenario of PCBA, but the performance of few-shot tasks of PCBA dropped significantly. One reason might be the different number of tasks used for training: the graph-related tasks involved in individual training are much more than those in joint training.

## F Limitations and future works

While OFA aims to provide a solution for the general graph foundation model, it is not yet able to perform regression tasks, because regression targets can be unbounded in values. Hence, if the range of all target values in a zero-shot regression task falls out of the range that OFA is trained on, it is very

Table 16: Few-shot results (Acc) on Cora (Node-level).

| # Way | 5-way | | | 2-way | | |
|---|---|---|---|---|---|---|
| Task | 5-shot | 1-shot | 0-shot | 5-shot | 1-shot | 0-shot |
| OFA-joint-lr | $48.76_{\pm2.65}$ | $34.04_{\pm4.10}$ | $28.72_{\pm9.90}$ | $76.10_{\pm4.11}$ | $67.44_{\pm4.47}$ | $56.92_{\pm3.09}$ |
| OFA-ind-lr | $42.28_{\pm2.35}$ | $31.28_{\pm2.63}$ | $23.68_{\pm1.67}$ | $72.20_{\pm3.82}$ | $62.22_{\pm1.17}$ | $51.85_{\pm4.35}$ |

Table 17: Results (AUC) on HIV and 2-way tasks.

| Task | 10-shot | 5-shot | 3-shot | 1-shot | 0-shot |
|---|---|---|---|---|---|
| OFA-joint-lr | 61.23±1.90 | 63.58±1.81 | 63.99±1.21 | 59.39±2.10 | 35.67±4.46 |
| OFA-ind-lr | 54.36±4.90 | 57.56±3.66 | 59.30±3.04 | 57.17±1.82 | 51.16±5.89 |

Table 18: Results (AUC) on PCBA and 2-way tasks

| Task | 10-shot | 5-shot | 3-shot | 1-shot | 0-shot |
|---|---|---|---|---|---|
| OFA-joint-lr | 51.64±9.90 | 51.53±9.94 | 51.58±9.59 | 51.72±8.57 | 60.62±5.45 |
| OFA-ind-lr | 54.58±2.90 | 54.80 ±3.75 | 54.67±4.35 | 54.92±4.38 | 52.71±5.57 |

difficult for OFA to predict the correct target value in that range, which is why we focus on general classification in this work. A potential approach is to specify a target range in the NOI prompt node task description, and let the model predict regression value based on the specified range. However, reasoning about math concepts is difficult even for most advanced LLMs, and certainly unreliable for our current adopted LLMs. Hence, we leave such an approach to future work.

While OFA is already trained in several different domains and the performance is on par or even outperforms some GNN and LLM approaches, the training data for the graph foundation model is still scarce compared to that of LLMs. Also, LLMs explore training techniques beyond supervised training, including auto-regressive training, and contrastive learning, which largely improve LLM's ability to model data and generalize to unseen tasks. Other unsupervised training techniques are possible, like the one proposed in GraphPrompt (Liu et al., 2023c). We believe these training techniques can further enhance the performance of OFA and regard this as an important direction to explore in the future.

