# OpenReview forum: "One For All: Towards Training One Graph Model For All Classification Tasks"
_ICLR.cc/2024/Conference — ICLR 2024 spotlight_

### Official Review · Reviewer_WF3H · 2023-10-16

**Soundness:** 3 good
**Presentation:** 4 excellent
**Contribution:** 3 good
**Rating:** 10
**Confidence:** 5

**Summary:**

The paper proposes One for All (OFA), a novel framework that addresses how to unify various graph tasks with various graph data.  OFA uses text-attributed graphs, allowing nodes and edges to be described in natural language, and introduces a new graph prompting method for diverse tasks without fine-tuning. When trained across multiple graph data domains, OFA demonstrates strong performance, making it a pioneering multi-purpose graph classification model.

**Strengths:**

1.LLM and graph prompting are both highly effective methods in their respective domains. Combining the two to address the cross-domain TAG problem is both convincing and well-motivated. I highly appreciate the idea of this paper, which is quite enlightening.
2. This paper is well-articulated.

**Weaknesses:**

1. A work on graph prompt learning should be cited and discussed.
* Tan, Zhen, et al. "Virtual Node Tuning for Few-shot Node Classification." arXiv preprint arXiv:2306.06063 (2023).
2. I highly recommend the authors to make the code public for readers to follow the work.
3. I suggest exploring non-meta-learning scenario such as in GraphPrompt.
* Liu, Zemin, et al. "Graphprompt: Unifying pre-training and downstream tasks for graph neural networks." Proceedings of the ACM Web Conference 2023. 2023.

**Questions:**

See Weaknesses.

---

> ### Author Response · Authors · 2023-11-14
> **Response to Reviewer WF3H**
>
> We sincerely thank reviewer WF3H for acknowledging our work and providing positive feedback. We address the reviewer’s concerns as follows.
>
> > W1: A work on graph prompt learning [1] should be cited and discussed.
> >
>
> We appreciate the recommendation to discuss this related work. We briefly discuss it here and will add it to our revision. VNT [1] offers a novel method for few-shot node classification (FSNC) by integrating trainable virtual node embeddings into the original graph, which serves as a form of graph prompting. For each new FSNC task, the pre-trained graph transformer model can be frozen and only the embedding of virtual nodes needs to be trained. However, VNT only focuses on node classification and the proposed virtual node needs to be retrained for each task. In contrast, OFA can perform node/link/graph few-shot classification without any retraining.
>
> > W2: I highly recommend the authors to make the code public for readers to follow the work.
> >
>
> Thanks for the suggestion! We will publicize our code in the future. Meanwhile, we have already provided an anonymized link to our code repository in the paper (in Appendix D). This repository contains the complete implementation details along with necessary documentation.
>
> > W3: I suggest exploring non-meta-learning scenario such as in GraphPrompt.
> >
>
> We thank the reviewer for raising this novel perspective and interesting work! During training for few-shot tasks, instead of constructing N-way K-shot tasks like meta learning, GraphPrompt [2] directly optimizes the task to similarize the embedding of nodes that are connected (each represented by node-centered subgraph) and dissimilarize those that are not connected. During the test, GraphPrompt determines the class of a target node by comparing its embedding with the prototype of each class.
>
> The key advantage of GraphPrompt lies in its unsupervised training approach, and we recognize the potential for our OFA framework to adopt a similar strategy with minimal adjustments. For instance, we could design pre-text tasks within OFA to predict the connection relationship between two Node of Interest (NOI) graphs. A straightforward implementation could be connecting two NOI graphs to the NOI prompt node. The raw text for the class nodes in this scenario could be sentences like 'these two nodes are connected' or 'these two nodes are not connected.’ Other contrastive methods can also be explored. We will add this discussion and related works in the revision and regard this as an important direction to explore in the future!
>
> References:
>
> [1] Tan et al., Virtual Node Tuning for Few-shot Node Classification, arXiv, 2023.
>
> [2] Liu et al., Graphprompt: Unifying pre-training and downstream tasks for graph neural networks, WWW, 2023.

---

> ### Author Response · Authors · 2023-11-19
>
> Dear Reviewer WF3H:
>
> As the discussion period comes to a close, we would like to thank you again for your positive assessment. Your support and inspirational comments have been invaluable to us. We remain open and eager to incorporate any further feedback or insights you might offer.

---

### Official Review · Reviewer_A8Qx · 2023-10-30

**Soundness:** 3 good
**Presentation:** 3 good
**Contribution:** 2 fair
**Rating:** 6
**Confidence:** 4

**Summary:**

This paper proposes One-for-All (OFA), a framework for building and training a single graph model that can perform various graph-related tasks across different domains. The experimental results on real-world datasets demonstrate the effectiveness and efficiency of the proposed framework.

**Strengths:**

1.	A unified graph model is designed for various tasks and different domains of graph data.

2.	The introduced model can encode features of different graphs into the same embedding space.

3.	Extensive experiments are conducted on various settings and datasets from different domains.

**Weaknesses:**

1.	The design of prompt nodes and utilizing NOI-subgraph to unify different tasks seem to be similar to several previous works. For example,  [1] introduced a learnable prompt vector to unify tasks, and [2-3] proposed to utilize super nodes to perform pooling readout. The proposed NOI prompt node appears to be a combination of these two lines of works. Could you include more in-depth discussions and comparisons with these related works.

2.	Comparison with methodologies that integrate LLMs and graph models, such as [4-7], is necessary.

3.	The Few-shot and Zero-shot results, especially for TLP-SURGL and TLP-BGRL, do not outperform baseline models. Could you include further discussions and analysis of these underwhelming performance results?

4. The ability of the LLMs to unify the feature space across datasets from diverse domains is very interesting. It would be insightful to see the unified feature representations of datasets within the same domain (e.g., two citation networks) and the datasets spanning different domains (e.g., a citation network and a molecular network). For example, some visual analysis will offer insights into the effectiveness of the LLM in harmonizing feature spaces across heterogeneous datasets.

5. Additional ablation studies can be conducted to see whether the constructed graph structure is helpful. For example, directly using the unified node features (produced by LLMs) as the input feature of some backbone GNN models like GIN, GCN, GAT.

6. The article introduces a method to employ LLMs to generate inputs for GNNs. Another line of works aims to utilize GNNs to generate inputs for LLMs, and it seems that this approach can better utilized LLMs' significant capabilities in addressing various problems on vast data. I'm curious about the difference between these two pipelines. Can you provide some explanations and compare the performance of these two pipelines.



[1] Zemin Liu, Xingtong Yu, Yuan Fang, and Xinming Zhang. Graphprompt: Unifying pre-training and downstream tasks for graph neural networks. In WWW, 2023.

[2] F. Hu, Y. Zhu, S. Wu, and T. Tan. Hierarchical graph convolutional networks for semi-supervised node classification. In IJCAI, 2019.

[3] Matthias Fey, Jan-Gin Yuen and Frank Weichert. Hierarchical Inter-Message Passing for Learning on Molecular Graphs. In ICML, 2020.

[4] Xiaoxin He, Xavier Bresson, Thomas Laurent, and Bryan Hooi. Explanations as features: Llm-based features for text-attributed graphs. arXiv preprint arXiv:2305.19523, 2023.

[5] Jianan Zhao, Meng Qu, Chaozhuo Li, Hao Yan, Qian Liu, Rui Li, Xing Xie, and Jian Tang. Learning on large-scale text-attributed graphs via variational inference. arXiv preprint arXiv:2210.14709, 2022.

[6] Ruosong Ye, Caiqi Zhang, Runhui Wang, Shuyuan Xu, and Yongfeng Zhang. Natural language is all a graph needs. arXiv preprint arXiv:2308.07134, 2023.

[7] Jianan Zhao, Le Zhuo, Yikang Shen, Meng Qu, Kai Liu, Michael Bronstein, Zhaocheng Zhu, and Jian Tang. Graphtext: Graph reasoning in text space. arXiv preprint arXiv:2310.01089, 2023.

**Questions:**

see weakness.

---

> ### Author Response · Authors · 2023-11-14
> **Response to Reviewer A8Qx Part 1/3**
>
> We greatly appreciate reviewer A8Qx for the insightful review to help us improve the paper. We address the reviewer’s concerns as follows:
>
> > W1: The design of prompt nodes and utilizing NOI-subgraph to unify different tasks seem to be similar to several previous works. For example, [1] introduced a learnable prompt vector to unify tasks, and [2-3] proposed to utilize super nodes to perform pooling readout. The proposed NOI prompt node appears to be a combination of these two lines of works. Could you include more in-depth discussions and comparisons with these related works.
> >
>
> We thank the reviewer for mentioning these works and some of the works are indeed inspirations to our paper. We briefly discuss them here and will include all discussions in the revision. Concretely, GraphPrompt [1] introduces a trainable vector to prompts on subgraphs for downstream tasks. This means that for a new task, it needs to train a new prompting vector, and hence it cannot perform zero-shot tasks. Whereas in our work, the prompting is through an NOI subgraph that is connected to the original graph. Particularly, task information is described in natural language and encoded by LLM as node features in the NOI subgraph. Thus, for a new class/task, we can describe the task by human-interpretable text and inject the information into the input graph to make predictions without any fine-tuning, like in the case of language models.
>
> We would like to emphasize the distinct roles of supernodes in H-GCN [2] and hierarchical inter-message passing [3] and the NOI prompt node in our work. Specifically, [2] and [3] both use a predefined non-neural algorithm to automatically organize nodes into groups pooled into supernodes and then perform message-passing on the supernodes. The algorithm generates fixed results based on the graph structure. The goal of these approaches is to use a pre-defined algorithm to better reason about particular structures in the graph. However, in our work, the connections between the NOI prompt node and nodes of interest are independent of the graph structure but user-defined and task-dependent. It is used to instruct the GNN to attend to the nodes that are the targets of the task.
>
> > W2: Comparison with methodologies that integrate LLMs and graph models, such as [4-7], is necessary.
> >
>
> We thank reviewer A8Qx for the suggestion to include more related works and briefly discuss them here. As this is a quickly emerging field, we were not able to include some of the mentioned concurrent work in the original submission [6, 7], but we will include all discussions in the revision. First, All of these methods [4-7] focus on node classification, while our method OFA trains one model for all the node-/link-/graph-level tasks across multiple datasets. Both ExpAsFeat [4] and GLEM [5] apply GNN to the graph with encoded text embedding generated by a Language Model (LM). ExpAsFeat uses LM to generate several different texts to better describe a node. GLEM aims to improve the text embedding quality by a variational EM algorithm that alternately updates GNN and LM. However, these methods require distinct training for different datasets. Both InstructGLM [6] and GraphText [7] transfer graph information into texts as the input to the LLM, with InstructGLM designing a prompt paradigm to describe node neighbors and GraphText proposing a graph-syntax tree to provide sorted sequences of nodes. These methods only describe graph connections by texts and do not use GNNs, thus they might lose essential graph structure information.
>
> > W3: The Few-shot and Zero-shot results, especially for TLP-SURGL and TLP-BGRL, do not outperform baseline models. Could you include further discussions and analysis of these underwhelming performance results?
> >
>
> We would like to first clarify that while TLP-SUGRL outperforms OFA in few-shot tasks, OFA outperforms TLP-BGRL on all tasks except the Cora 5-shot task. Note that OFA is tested in a transfer setting on the Cora dataset: the Cora dataset is not included in the training set, while TLP-BGRL and TLP-SUGRL are both trained and tested on the Cora dataset. Also, neither TLP-SUGRL nor TLP-BGRL can perform zero-shot tasks, while OFA can.
>
> The outstanding performance of TLP-SUGRL, as pointed out by recent research [8], can be attributed to its unsupervised contrastive training paradigm which also includes test data during training (no label information is leaked, but test node information is present during contrastive learning). Such a paradigm is valid for transductive learning settings, where training and test nodes are on the same graph, but will fail in inductive settings, where training and test nodes are not on the same graph. Conversely, OFA still works on inductive settings.

---

> ### Author Response · Authors · 2023-11-14
> **Response to Reviewer A8Qx Part 2/3**
>
> > W4: The ability of the LLMs to unify the feature space across datasets from diverse domains is very interesting. It would be insightful to see the unified feature representations of datasets within the same domain (e.g., two citation networks) and the datasets spanning different domains (e.g., a citation network and a molecular network). For example, some visual analysis will offer insights into the effectiveness of the LLM in harmonizing feature spaces across heterogeneous datasets.
> >
>
> Thanks for your insightful suggestion! We provide visualization of generated node embedding for different datasets in (https://anonymous.4open.science/r/OFA-6745/visualization_dataset.pdf) and will include it in the revision. We can see that LLM indeed embeds nodes from different domain to different space, which further explain the reason why a single GNN can work effectively for data from different domain using the OFA framework.
>
> > W5: Additional ablation studies can be conducted to see whether the constructed graph structure is helpful. For example, directly using the unified node features (produced by LLMs) as the input feature of some backbone GNN models like GIN, GCN, GAT.
> >
>
> We would like to first highlight the ablation study in Table 12. We conducted experiments on ArXiv and HIV datasets. The experiment includes three cases: 1, Full, which uses the proposed NOI prompt graph. 2, -NOI prompt node, which connects class nodes directly to the NOI. 3, *(The case that the reviewer kindly suggested)* -Class node or no prompting, which has no prompt node and feeds the original graph to a GNN. In this case, to make a prediction, we apply mean pooling to the NOI, concatenate the pooled vector with the class description embedding vector, and apply MLP to the final vector for binary classification.
>
> All three cases can either **jointly** or **separately** learn both ArXiv and HIV datasets. We compare their performance under these two cases. We observe that the proposed graph structure does not help with learning when datasets are trained separately because the downstream MLP only needs to learn one prediction target, and the proposed structure is essentially pooling on the NOI. However, we also observe that when the two datasets are trained together, the proposed structure **considerably increases the performance** over other cases. In joint training, as the downstream MLP needs to learn multiple tasks, it will be very difficult for the MLP to switch to corresponding tasks based only on a pooled vector. On the contrary, the proposed graph structure helps to distinguish different tasks for the MLP, because text-based task description is provided as the prompt node’s feature.
>
> We additionally conducted an experiment comparing *cases of Full and -Class node* using the same setup but with more datasets, and show the results below. Note we use GCN as the backbone model.
>
> |  | Joint/Full | Joint/-Class node | Separate/Full | Separate/-Class node |
> | --- | --- | --- | --- | --- |
> | ArXiv | 75.23±0.03 | 75.19±0.10 | 75.06±0.08 | 75.39±0.09 |
> | HIV | 75.92±0.45 | 71.60±0.54 | 75.81±0.36 | 75.43±0.50 |
> | Cora-node | 75.48±0.29 | 71.39±0.07 | 75.72±0.49 | 76.12±0.87 |
> | Cora-link | 92.27±0.84 | 89.51±0.46 | 93.16±0.95 | 92.80±0.62 |
>
> The models show similar patterns when there are more datasets. We notice that the joint training performance on three out of the four datasets significantly drops when we switch from the proposed prompting to no prompting. Besides the performance consideration, the design of graph prompting is to unify different tasks for in-context learning. For example, we need to significantly alter the model to perform few-/zero-shot learning using the no prompting approach, while the current approach does not need to modify any component of the trained model.

---

> ### Author Response · Authors · 2023-11-14
> **Response to Reviewer A8Qx Part 3/3**
>
> > W6: The article introduces a method to employ LLMs to generate inputs for GNNs. Another line of works aims to utilize GNNs to generate inputs for LLMs, and it seems that this approach can better utilized LLMs' significant capabilities in addressing various problems on vast data. I'm curious about the difference between these two pipelines. Can you provide some explanations and compare the performance of these two pipelines.
> >
>
> We thank the reviewer again for this thoughtful comment. We did some research and didn’t find an exact match to the reviewer’s suggestion “utilize GNNs to generate inputs for LLMs”. But we did find related research that uses graph descriptive language to transform a graph into a text sequence, and directly apply an LLM to the sequence to make downstream predictions. (If this is not what the reviewer meant, please let us know in the comment.) For a detailed comparison between using LLM as a graph feature enhancer versus as a predictor, we kindly refer the reviewer to our general response.
>
> These works are certainly influential. Their most prominent advantage is that they are naturally integrated into LLM. Consequently, it is easy to adapt it to have conversational/explanatory abilities. However, these approaches only minimally incorporate graph topological information which is the core of using graph data, because the connections are described by plain texts and such description is difficult for the LLM to learn as pointed out by preliminary works like [9]. Note that most of these works also extract subgraphs around prediction targets first and use the graph descriptive language to describe the subgraph. The LLM might only serve as a summarizer of the neighborhood information rather than learning the structure. On the contrary, in OFA, the graph structures are explicitly preserved. Recent development on GNN shows that GNN is a reliable architecture for learning graph structure which guarantees that important topological information is not lost.
>
> References:
>
> [1] Liu et al., Graphprompt: Unifying pre-training and downstream tasks for graph neural networks, WWW, 2023.
>
> [2] Hu et al., Hierarchical graph convolutional networks for semi-supervised node classification, IJCAI, 2019.
>
> [3] Fey et al., Hierarchical Inter-Message Passing for Learning on Molecular Graphs, ICML, 2020.
>
> [4] He et al., Explanations as features: Llm-based features for text-attributed graphs, arXiv, 2023.
>
> [5] Zhao et al., Learning on large-scale text-attributed graphs via variational inference, arXiv 2022.
>
> [6] Ye et al., Natural language is all a graph needs, arXiv, 2023.
>
> [7] Zhao et al., Graphtext: Graph reasoning in text space, arXiv, 2023.
>
> [8] Liu et al., Graph Contrastive Learning Meets Graph Meta Learning: A Unified Method for Few-shot Node Tasks, arXiv, 2023.
>
> [9] Guo et al. GPT4Graph: Can Large Language Models Understand Graph Structured Data? An Empirical Evaluation and Benchmarking, arXiv, 2023.

---

> ### Author Response · Authors · 2023-11-19
>
> Dear Reviewer A8Qx:
>
> We sincerely appreciate your detailed and constructive review of our paper. As the discussion period is near its end, we would like to ensure our response aligns with your expectations and addresses your concerns. In our response, we have discussed the uniqueness of our method OFA compared to related works and clarified our few-shot and zero-shot results. We also provided additional visualization and experiments to better demonstrate the model design. We appreciate your feedback and look forward to any further comments.

---

> ### Author Response · Authors · 2023-11-22
>
> Dear Reviewer A8Qx:
>
> We just want to reach out to you again and see if our response addresses your concern. Your comments really inspire us, and we are eager to continue discussing our work with you.

---

> > ### Comment · Reviewer_A8Qx · 2023-11-23
> > **Thanks for your response**
> >
> > Thank you for providing comprehensive responses regarding additional results and discussions. The majority of my previous concerns have been addressed. Consequently, I will raise my score from 5 to 6.

---

### Official Review · Reviewer_RxxK · 2023-11-01

**Soundness:** 3 good
**Presentation:** 3 good
**Contribution:** 3 good
**Rating:** 6
**Confidence:** 4

**Summary:**

In the field of artificial intelligence, creating a single model to handle diverse tasks has been a longstanding goal. Large language models have excelled in language-related tasks, but applying this versatility to graph-based tasks is challenging. Graph data from different domains have unique attributes and distributions, making uniform representation difficult. Additionally, graph tasks encompass nodes, links, and graphs, necessitating distinct strategies. Furthermore, context-aware learning for graphs lacks a clear method.

To address these challenges, this paper introduced "One for All" (OFA), a framework that uses a single graph model to conquer these obstacles. OFA uses text-attributed graphs, unifying diverse graph data and standardizing task representation with "nodes-of-interest." It pioneers a novel graph prompting approach for versatile task handling without fine-tuning. They train OFA using data from various domains and evaluate its performance in diverse learning scenarios. The results establish OFA as the first general-purpose graph classification model, revolutionizing the field of graph-based artificial intelligence.

**Strengths:**

The motivation of this paper is well-delivered and interesting

The proposed method is novel and seems to be good according to their experimental report

**Weaknesses:**

W1. translating graph data with text feature sequences might lose some key information from graphs.

W2. According to Figure 1, LLM enhances text for text-attributed graphs. it still needs a GNN as a predictor. (1) I wonder whether the GNN model is also pre-trained and frozen. and (2) what are the results compared with using LLM as a predictor instead of an enhancer?

W3. It is unclear how to use NOI prompt nodes and class nodes with GNN to predict the downstream tasks. according to Figure 2, can I say that the downstream tasks (node, edge, graph classifications) are treated as predicting links between NOI prompt and class nodes? It seems that NOI prompt nodes can be treated as a special case of the prompt graph mentioned in the paper "All in One" (Sun et al., 2023), what's the differences between them? and why the authors use only one NOI prompt node instead of multiple NOI prompt nodes.

W4. I wonder why the authors use sentence transformer instead of ChatGPT API, or LLAMA, etc since they claim that "any kind of LLM can be used as the encoder, and a stronger LLM potentially yields better overall performance." Is it possible that a very large language model contains too much unrelated knowledge/intelligence that may reduce task performance?

W5. I wonder what would happen if you Re-order the item in your prompt for ChatGPT.  It seems the reason why you use sentence transformer is that it is not sensitive to the order of your prompt? if you change to ChatGPT, different orders may generate entirely different sentences, making the downstream performance unpredictable

**Questions:**

see W1-4

I would like to see the rebuttal to the questions mentioned in the above section “Paper Weakness”. I’m afraid that I might have not sufficient time to see a very long rebuttal. A concise and clear one would be good.

The potential weakness won't prevent me from raising my final score. I just want to make clear whether my understanding is correct.

---

> ### Author Response · Authors · 2023-11-14
> **Response to Reviewer RxxK**
>
> We thank reviewer RxxK for detailed and knowledgeable feedback, and we address the reviewer’s concerns as follows. Per the reviewer’s request, we tried to keep the response succinct. We are happy to explain further if the reviewer’s concerns remain after the response.
>
> > W1: translating graph data with text feature sequences might lose some key information from graphs.
> >
>
> Our method uses LLM to only encode text that exists in graphs (node and edge attributes) from different domains to the same vector embedding space. The embedding vectors then serve as original node/edge features in graphs for GNN to predict. The graph structure/adjacency information is intact. Therefore, our method can maintain both graph structure and feature information.
>
> > W2: According to Figure 1, LLM enhances text for text-attributed graphs. it still needs a GNN as a predictor. (1) I wonder whether the GNN model is also pre-trained and frozen. and (2) what are the results compared with using LLM as a predictor instead of an enhancer?
> >
>
> (1) For both supervised and few-shot scenarios, we first train our randomly initialized GNN model on the training data using the label information. Then, our GNN model is frozen for any new tasks. (2) We provide an additional comparison with baselines using LLM as a predictor in the general response. Key findings show that these models often excel in node classification but may overlook graph structure. In contrast, OFA maintains graph topology using GNNs and is capable of handling multiple task levels across various datasets simultaneously.
>
> > W3: It is unclear how to use NOI prompt nodes and class nodes with GNN to predict the downstream tasks. according to Figure 2, can I say that the downstream tasks (node, edge, graph classifications) are treated as predicting links between NOI prompt and class nodes? It seems that NOI prompt nodes can be treated as a special case of the prompt graph mentioned in the paper "All in One" (Sun et al., 2023), what's the differences between them? and why the authors use only one NOI prompt node instead of multiple NOI prompt nodes.
> >
>
> **Short response:** (1) We apply an MLP binary classifier to the embedding of the class nodes generated from a GNN to make classifications. (2) The embedding of NOI prompt nodes encodes task information with human-readable texts, which endows OFA with zero-shot ability. Whereas, the prompt graph in All-in-One [1] is obtained through task-specific training, which needs to be retrained once facing new tasks. (3) The NOI prompt node is used to integrate both task and graph information and one NOI prompt node with our designed connection pattern is both effective and simple.
>
> **Additional discussion:** Here we provide a detailed discussion on point (1). All nodes in the prompted graph, including the NOI prompt nodes and class nodes, have text descriptions. The texts of NOI nodes are task descriptions, such as “node classification on literature category” and “graph classification on molecule property”. The texts of class nodes are descriptions of specific classes for a task described in the NOI prompt node, such as “Computer science literature category” and “HIV inhibition molecule property”. We use LLMs to convert texts to node vector features, and the graphs with vector features are fed to GNN to generate node vector representations. The vector representations of the class nodes contain **task, class, and graph information** because of the GNN. We then apply MLP to the class nodes to make binary classification and select the class with the highest probability.
>
> > W4: I wonder why the authors use sentence transformer instead of ChatGPT API, or LLAMA, etc. Is it possible that a very large language model contains too much unrelated knowledge/intelligence that may reduce task performance?
> >
>
> We chose a smaller version of LLM for two reasons: (1) It is easier to work with, which allows flexible deployment and development. (2) As the reviewer’s perceptive question suggested, concurrent work [2] shows that larger LLMs such as ChatGPT and LLaMa are specifically tuned for conversational purposes, and might not perform as well in sentence encoding scenarios.
>
> > W5: I wonder what would happen if you Re-order the item in your prompt for ChatGPT. It seems the reason why you use sentence transformer is that it is not sensitive to the order of your prompt?
> >
>
> Our method uses LLM to only encode text that exists in graphs (node and edge attributes)  to the same vector embedding space. Since each node and edges are encoded independently, our encoding schema is invariant to the order of nodes and edges.
>
> Reference:
>
> [1] Sun et al., All in One: Multi-task Prompting for Graph Neural Networks, KDD, 2023.
>
> [2] Chen et al., Exploring the Potential of Large Language Models (LLMs) in Learning on Graphs, arXiv, 2023.

---

> ### Author Response · Authors · 2023-11-19
>
> Dear Reviewer RxxK:
>
> Thank you again for acknowledging our contribution and raising insightful points to improve our paper. As the discussion period ends soon, we would like to check whether our response addresses your concerns. Our explanations have provided clarity on the NOI prompt structure, the training procedure, and the comparison between models using LLM as a predictor. Looking forward to your feedback!

---

### Official Review · Reviewer_LMu3 · 2023-11-09

**Soundness:** 2 fair
**Presentation:** 3 good
**Contribution:** 3 good
**Rating:** 6
**Confidence:** 3

**Summary:**

This paper proposes a unified paradigm for learning different tasks over graphs and across different domains. Notably, the proposed algorithms achieve competitive performance in various tasks.

**Strengths:**

1. The framework proposed by the author is very interesting. The idea of prompted nodes naturally unifies the three major tasks in one simple but powerful problem formulation.
2. The presentation of the result is clear with both quantitative and qualitative results.

**Weaknesses:**

1. The soundness of the experiments seems lacking. Based on my understanding, LLM is one if the key component to achieve cross-task and cross-domain generalization. However, the choice of LLM is rarely discussed. I didn't find detailed description of the LLM used in the main experiments. Neither a comparison between different LLMs is missing in main paper and appendix.
2. The modeling of link-level task lack details. For example, do you have two or one class node?

**Questions:**

Please refer to the weakness for questions.

---

> ### Author Response · Authors · 2023-11-14
> **Response to Reviewer LMu3**
>
> We greatly appreciate reviewer LMu3’s insightful feedback and critical comments to help us refine our work. We address the reviewer’s concerns as follows:
>
> > W1: The soundness of the experiments seems lacking. Based on my understanding, LLM is one if the key component to achieve cross-task and cross-domain generalization. However, the choice of LLM is rarely discussed. I didn't find detailed description of the LLM used in the main experiments. Neither a comparison between different LLMs is missing in main paper and appendix.
> >
>
> We have mentioned the use of sentence transformer as our LLM at the end of section 3.1, and we will give a more detailed discussion regarding the choice of the LLM in our revision. As reviewer LMu3 acutely pointed out, LLM is indeed an indispensable part of our framework, but the framework is not limited to any particular LLM model. Rather, the key contribution of the paper is that we can use any LLM to unify cross-domain graph tasks, and we chose the LLM that is easy to work with to demonstrate the validity of such an approach.
>
> Furthermore, we agree with reviewer LMu3 that a study of the effect of LLM in our model is important to evaluate the framework. Hence, we provide additional experimental results. We select another LLM e5-large-v2 [1] (will include results for Llama2-7B and Llama2-13B [2] once we have it) and conduct experiments on the supervised learning scenario using all datasets (same experimental setting as OFA-joint in the paper). The results are shown in the following:
>
> | Dataset | Cora  | Cora  | PubMed  | PubMed  | ogbn-arxiv  | Wiki-CS | FB15K237 | WN18RR | HIV | PCBA |
> | --- | --- | --- | --- | --- | --- | --- | --- | --- | --- | --- |
> | Task Type | Link | Node | Link | Node | Node | Node | Link | Link | Graph | Graph |
> | Metric | AUC ↑ | Acc ↑ | AUC ↑ | Acc ↑ | Acc ↑ | Acc ↑ | Acc ↑ | Acc ↑ | AUC ↑ | APR ↑ |
> | OFA-joint-ST | 94.04±0.49 | 75.90±1.26 | 98.21±0.02 | 75.54±0.05 | 75.54±0.11 | 78.34±0.35 | 95.54±0.06 | 96.91±0.11 | 78.02±0.17 | 24.83±0.10 |
> | OFA-joint-e5 | 92.75±0.42 | 70.73±1.69 | 98.44±0.06 | 78.09±1.71 | 75.81±0.11 | 72.51±0.39 | 95.27±0.35 | 97.76±0.39 | 78.56±1.69 | 25.30±0.29 |
>
> From the result, we can see that our framework still works effectively with different LLMs. Moreover, we can see that different LLMs encode language differently which results in slightly different performance on individual datasets.
>
> > W2: The modeling of link-level task lack details. For example, do you have two or one class node?
> >
>
> The number of class nodes depends on the number of target link classes.
>
> **For homogeneous graphs**, where the goal is to predict whether a link exists between two nodes, we have one class node. Since we perform binary classification using class node embedding, a prediction result of 1 indicates that a link exists between the two nodes, and a result of 0 indicates the opposite. For example, in a co-citation network where links represent co-citations, the goal is to predict if two papers are co-cited. Then for two target papers, we connect a NOI prompt node to the two papers’ corresponding nodes and connect one class node to the NOI prompt node. The text of the class node is “The papers are co-cited”. Then, if the binary classification result gives 1, the model predicts the link exits, and if the result gives 0, the model predicts the link does not exist. **For heterogeneous graphs/knowledge graphs**, we can have more class nodes because two nodes can be connected by different relations, like “lives in”, “parent of”, and “works for”. The construction of the prompt graph is similar. Both nodes in the link connect to the NOI prompt node, and the NOI prompt node connects to k class nodes, where k is the number of target relations. The binary classification results on class nodes represent the likelihood of relations corresponding to the class nodes. Note that link-level tasks are very similar to node-level and graph-level tasks and can be unified into the same graph-prompting framework introduced in the paper. They only differ by their nodes of interest (NOI), and hence their connections between the NOI and the NOI prompt nodes are different.
>
> Reference:
>
> [1] Wang et al., Text Embeddings by Weakly-Supervised Contrastive Pre-training, arXiv 2022.
>
> [2] Touvron et al., Llama 2: Open Foundation and Fine-Tuned Chat Models, arXiv 2023.

---

> > ### Comment · Reviewer_LMu3 · 2023-11-22
> > **Thanks for the additional results**
> >
> > Thanks for author's clarification on link prediction and different backbone LMs. It would be nice if you can consider more recent LLMs in your results to demonstrates the applicability of the proposed framework. As a result, I would like to raise my score.

---

> > > ### Author Response · Authors · 2023-11-22
> > >
> > > Thanks for your constructive comments again and for raising your score! We will include more recent LLMs in our revised paper to better support our evaluation.

---

> ### Author Response · Authors · 2023-11-19
>
> Dear Reviewer LMu3:
>
> As the discussion period ends soon, we would like to check whether our responses answer your questions. Following your comments, we conducted experiments to test different choices of LLM and provided a detailed explanation of the model to address your concerns. Thank you again for your comments and suggestions to improve our paper, and we look forward to your reply.

---

### Author Response · Authors · 2023-11-14
**General Response to All Reviewers**

Dear Reviewers,

We sincerely appreciate the time and effort you have dedicated to reviewing our paper. Your insight is invaluable in refining our work. Many of your positive feedbacks inspired us, such as,

1. The novelty of the Unified Learning Paradigm: We are sincerely thankful for the reviews describing our work as “very interesting” (Reviewer LMu3, RxxK), “novel” (Reviewer RxxK, WF3H), “enlightening” (Reviewer WF3H), and even “revolutionary” (Reviewer RxxK). We also appreciate the reviewer’s acknowledgment of our contribution, including “unifying major tasks” in graph learning (Reviewer LMu3, RxxK, A8Qx) and “addressing cross-domain TAG tasks” (Reviewer WF3H).
2. Sound Presentation and Extensive Experiments: We are also glad that we delivered the ideas and essence of our work successfully to the readers, as the reviewers agreed that our paper is “well-delivered” (Reviewer RxxK) and “well-articulated” (Reviewer WF3H). Moreover, multiple reviewers (Reviewer LMu3, Rxxk, A8Qx) seemed to agree we conducted extensive experiments to validate our approach.

These uplifting comments not only acknowledged our work but also encouraged us to refine the paper.

The reviewer's concerns and questions also spurred us to dig deeper into the topic, and here we address some common concerns based on feedback. The point-to-point response to all concerns raised by each reviewer is self-contained.

> Discussion on LLMs in Graph Modeling: Predictor vs. Feature Enhancer.
>

Our model, OFA, employs LLMs as a feature enhancer to unify node and edge features, with GNNs as predictors. This approach contrasts with recent methodologies that utilize texts to describe both graph features and structure, employing LLMs as predictors. We appreciate Reviewer Rxxk and A8Qx's suggestion for a detailed comparison between these two kinds of methods.

To this end, we compare OFA with five baseline models that use LLMs as predictors. Models like ExpAsFeat [1], InstructGLM [2], GraphText [3], and GraphGPT [4] focus on node classification, converting node features and substructures into text for LLM processing. GIMLET [5] focuses on zero-shot graph classification by using SMILE molecule sequences to describe the molecular graph. Among them, [2], [3], [4] are considered concurrent work according to the ICLR guideline.

Our comparative results are summarized in the following table. Note that 'NA' indicates the model's incapacity for a specific task, while '*' denotes that the model is capable of the task but uses different data splits or does not provide results.

|  | ogbn-arxiv | Cora-node | PubMed-node | cora-zeroshot | pcba-zeroshot |
| --- | --- | --- | --- | --- | --- |
| ExpAsFeat[1] | 75.20±0.03 | * | * | NA | NA |
| InstructGLM[2] | 75.70±0.12 | * | 89.6±0.4 | NA | NA |
| GraphText[3] | * | 76.5 | * | NA | NA |
| GraphGPT[4] | 75.11 | * | * | * | NA |
| GIMLET[5] | NA | NA | NA | NA | 62.11 |
| OFA-joint | 75.54±0.11 | 75.90±1.26 | 75.54±0.05 | 28.72±9.09 | 60.62±5.45 |

Current works employing LLM as a predictor mostly focuses on a single task level (node or graph), while OFA trains one graph model for various tasks. These baselines show advantages in node classification tasks, primarily because such tasks often involve categorizing a paper based on its title and abstract, and LLMs excel at understanding and contextualizing language. However, for link-/graph-level tasks that rely more on understanding the graph structure, using texts to describe graph topological connections may lead to a subpar understanding of critical structural information and significantly increase the learning difficulty of LLMs due to the extra long token sequence. In this context, LLMs often function as summarizers of local neighbor information, potentially overlooking the graph structure which is much more complex than neighbor/degree information. In contrast, our model OFA preserves graph structures explicitly by using GNNs. This difference highlights OFA’s strength in utilizing the topology of graph data.

References:

[1] He et al., Explanations as Features: Llm-based Features for Text-attributed Graphs, arXiv, 2023.

[2] Ye et al., Natural Language is All a Graph Needs, arXiv, 2023.

[3] Zhao et al., GraphText: Graph Reasoning in Text Space, arXiv, 2023.

[4] Tang et al., GraphGPT: Graph Instruction Tuning for Large Language Models, arXiv, 2023.

[5] Zhao et al., GIMLET: A Unified Graph-Text Model for Instruction-Based Molecule Zero-Shot Learning, NeurIPS 2023.

---

### Meta-Review · Area_Chair_kc9r · 2023-12-15

**Metareview:**

This paper presents an ambitious framework that leverages LLMs to handle various graph learning tasks using one model. The proposed method is well-motivated, and also demonstrates reasonably good performance. Especially, the ability to achieve non-trivial generalization for 0-shot settings is impressive. All reviewers were positive about this work.

**Justification For Why Not Higher Score:**

The model performance still underperforms state-of-the-art specialist methods for each individual task. There is room to further improve the proposed method to make it replace existing specialist solutions.

**Justification For Why Not Lower Score:**

The proposed framework offers a promising direction for developing general models for graph learning.

---

### Decision · Program_Chairs · 2024-01-16

Accept (spotlight)